



The ABCflux database: Arctic-Boreal CO$_2$ flux
observations and ancillary information aggregated to
monthly time steps across terrestrial ecosystems
Authors: Anna-Maria Virkkala[1], Susan M. Natali[1], Brendan M. Rogers[1], Jennifer D. Watts[1],
Kathleen Savage[1], Sara June Connon[1], Marguerite Mauritz[2], Edward A.G. Schuur[3], Darcy Peter[1],
Christina Minions[1], Julia Nojeim[1], Roisin Commane[4], Craig A. Emmerton[5], Mathias Goeckede[6],
Manuel Helbig[7,8], David Holl[9], Hiroki Iwata[10], Hideki Kobayashi[11], Pasi Kolari[12], Efrén López-
Blanco[13,14], Maija E. Marushchak[15,16], Mikhail Mastepanov[14,17], Lutz Merbold[18], Frans-Jan W.
Parmentier[19,20], Matthias Peichl[21], Torsten Sachs[22], Oliver Sonnentag[8], Masahito Ueyama[23],
Carolina Voigt[15,8], Mika Aurela[24], Julia Boike[25,26], Gerardo Celis[27], Namyi Chae[28], Torben R.
Christensen[14], M. Syndonia Bret-Harte[29], Sigrid Dengel[30], Han Dolman[31], Colin W. Edgar[29], Bo
Elberling[32], Eugenie Euskirchen[29], Achim Grelle[33], Juha Hatakka[24], Elyn Humphreys[34], Järvi
Järveoja[21], Ayumi Kotani[35], Lars Kutzbach[9], Tuomas Laurila[24], Annalea Lohila[24,12], Ivan
Mammarella[12], Yojiro Matsuura[36], Gesa Meyer[8,37], Mats B. Nilsson[21], Steven F. Oberbauer[38],
Sang-Jong Park[39], Roman Petrov[40], Anatoly S. Prokushkin[41], Christopher Schulze[8,42], Vincent L.
St.Louis[5], Eeva-Stiina Tuittila[43], Juha-Pekka Tuovinen[24], William Quinton[44], Andrej Varlagin[45],
Donatella Zona[46], Viacheslav I. Zyryanov[41]
Woodwell Climate Research Center, 149 Woods Hole Road Falmouth, MA, 02540-1644, USA
University of Texas, at El Paso, 500 W University Rd, El Paso, TX 79902, USA
Center for Ecosystem Science and Society, and Department of Biological Sciences, Northern
Arizona University, Flagstaff, AZ, 86001
Dept. of Earth & Environmental Sciences, Lamont-Doherty Earth Observatory, Columbia
University, Palisades, NY 10964
Department of Biological Sciences, University of Alberta, Edmonton, Alberta, Canada T6G
2E9
Dept. Biogeochemical Signals, Max Planck Institute for Biogeochemistry, Jena, Germany
Department of Physics and Atmospheric Science, Dalhousie University, Halifax, Nova Scotia,
Canada
Departement de Geographie, Universite de Montreal, Montreal, Quebec, Canada





Institute of Soil Science, Center for Earth System Research and Sustainability (CEN), Universität Hamburg, Hamburg, Germany
Department of Environmental Science, Shinshu University, Matsumoto, Japan
Research Institute for Global Change, Japan Agency for Marine-Earth Science and Technology, Yokohama, Japan
Institute for Atmospheric and Earth System Research/Physics, Faculty of Science, University of Helsinki, Finland
Department of Environment and Minerals, Greenland Institute of Natural Resources, Kivioq 2, 3900, Nuuk, Greenland
Department of Bioscience, Arctic Research Center, Aarhus University, Frederiksborgvej 399, 4000 Roskilde, Denmark
Department of Environmental and Biological Sciences, University of Eastern Finland, Kuopio, Finland
Department of Biological and Environmental Science, University of Jyväskylä, Jyväskylä, Finland
Oulanka research station, University of Oulu, Liikasenvaarantie 134, 93900 Kuusamo, Finland
Agroscope, Research Division Agroecology and Environment, Reckenholzstrasse 191, 8046 Zurich, Switzerland
Center for Biogeochemistry in the Anthropocene, Department of Geosciences, University of Oslo, 0315 Oslo, Norway
Department of Physical Geography and Ecosystem Science, Lund University, 223 62 Lund, Sweden
Department of Forest Ecology and Management, Swedish University of Agricultural Sciences, 901 83 Umeå, Sweden
GFZ German Research Centre for Geosciences, Telegrafenberg, Potsdam, Germany
Graduate School of Life and Environmental Sciences, Osaka Prefecture University, 1-1 Gakuencho, Naka-ku, Sakai, 599-8531, Japan
Finnish Meteorological Institute, Climate system research, Helsinki, Finland
Alfred Wegener Institute Helmholtz Center for Polar and Marine Research, Telegrafenberg A45, 14473 Potsdam, Germany & Geography Department, Humboldt-Universität zu Berlin, Unter den Linden 6, 10099 Berlin, Germany
Geography Department, Humboldt-Universität zu Berlin, Berlin, Germany
Agronomy Department, University of Floria, Gainesville, USA
Institute of Life Science and Natural Resources, Korea University, 145 Anam-ro,Seongbuk-gu, Seoul, 02841, Republic of Korea
Institute of Arctic Biology, University of Alaska Fairbanks, Fairbanks, AK 99775, USA
Earth and Environmental Sciences Area. Lawrence Berkeley National Lab, Berkeley, CA 94720, USA
Department of Earth Sciences, VU University of Amsterdam, Amsterdam, The Netherlands
Center for Permafrost, Department of Geosciences and Natural Resource Management, University of Copenhagen, Øster Voldagde 10
Department of Ecology, Swedish University of Agricultural Sciences, Uppsala
Department of Geography & Environmental Studies, Carleton University, 1125 Colonel By Dr. Ottawa, ON, K2B 5J5 Canada
Graduate School of Bioagricultural Sciences, Nagoya University, Nagoya, Japan





Forestry and Forest Products Research Institute
Environment and Climate Change Canada, Climate Research Division, Victoria, BC V8N
1V8, Canada
Department of Biological Sciences and Institute of Environment, Florida International
University, Miami Florida 33199 USA
Division of Atmospheric Sciences, Korea Polar Research Institute, 26 Sondgomirae-ro
Yeonsu-gu, Incheon, Republic of Korea 21990
Institute for Biological Problems of Cryolithozone of the Siberian Branch of the RAS -
Division of Federal Research Centre "The Yakut Scientific Centre of the Siberian Branch of the
Russian Academy of Sciences
VN Sukachev Institute of forest SB RAS, Akademgorodok 50/28, Krasnoyarsk 660036
Russia
Department of Renewable Resources, University of Alberta, Edmonton, Alberta, Canada
T6G 2E9
School of Forest Sciences, University of Eastern Finland, Finland
Cold Regions Research Centre, Wilfrid Laurier University, Waterloo, Ontario, Canada, N2L
3C5
45 A.N. Severtsov Institute of Ecology and Evolution, Russian Academy of Sciences, 119071,
Leninsky pr.33, Moscow, Russia
Department of Biology, San Diego State University
ORCID iDs:
AMV: 0000-0003-4877-2918
SMN: 0000-0002-3010-2994
BMR: 0000-0001-6711-8466
MaM: 0000-0001-8733-9119
EAGS: 0000-0002-1096-2436
RC: 0000-0003-1373-1550
MG: 0000-0003-2833-8401
MH: 0000-0003-1996-8639
DH: 0000-0002-9269-7030
HI: 0000-0002-8962-8982
HK: 0000-0001-9319-0621
PK: 0000-0001-7271-633X
ELB: 0000-0002-3796-8408
MEM: 0000-0002-2308-5049
MiM: 0000-0002-5543-0302
LM: 0000-0003-4974-170X
FJP: 0000-0003-2952-7706
MP: 0000-0002-9940-5846
TS: 0000-0002-9959-4771
MU: 0000-0002-4000-4888
CV: 0000-0001-8589-1428
MA: 0000-0002-4046-7225
JB: 0000-0002-5875-2112
GC: 0000-0003-1265-4063



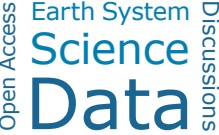

TRC: 0000-0002-4917-148X
MSB: 0000-0001-5151-3947
SD: 0000-0002-4774-9188
CE: 0000-0002-7026-8358
BE: 0000-0002-6023-885X
SEE: 0000-0002-0848-4295
AG: 0000-0003-3468-9419
EH: 0000-0002-5397-2802
JJ: 0000-0001-6317-660X
AK: 0000-0003-0350-0775
LK: /0000-0003-2631-2742
TL: 0000-0002-1967-0624
AL: 0000-0003-3541-672X
IM: 0000-0002-8516-3356
GM: 0000-0003-3199-5250
MBN: 0000-0003-3765-6399
SFO: 0000-0001-5404-1658
SJP: 0000-0002-6944-6962
RP: 0000-0002-6877-3902
ASP: 0000-0001-8721-2142
CS: 0000-0002-6579-0360
VLStL: 0000-0001-5405-1522
EST: 0000-0001-8861-3167
JPT: 0000-0001-7857-036X
WQ: 0000-0001-5707-4519
DZ: 0000-0002-0003-4839
VIZ: 0000-0002-1748-4801
AV: 0000-0002-2549-5236
CAE: 0000-0001-9511-9191
Word count: 9200 (abstract: 330, main text: 6400)
Corresponding author: Anna-Maria Virkkala, avirkkala@woodwellclimate.org






Abstract
Past efforts to synthesize and quantify the magnitude and change in carbon dioxide ($CO_2$) fluxes in
terrestrial ecosystems across the rapidly warming Arctic-Boreal Zone (ABZ) have provided valuable
information, but were limited in their geographical and temporal coverage. Furthermore, these efforts
have been based on data aggregated over varying time periods, often with only minimal site ancillary
data, thus limiting their potential to be used in large-scale carbon budget assessments. To bridge these
gaps, we developed a standardized monthly database of Arctic-Boreal $CO_2$ fluxes (ABCflux) that
aggregates *in-situ* measurements of terrestrial net ecosystem $CO_2$ exchange and its derived partitioned
component fluxes: gross primary productivity and ecosystem respiration. The data span from 1989 to
2020 with over 70 supporting variables that describe key site conditions (e.g., vegetation and disturbance
type), micrometeorological and environmental measurements (e.g., air and soil temperatures) and flux
measurement techniques. Here, we describe these variables, the spatial and temporal distribution of
observations, the main strengths and limitations of the database, and the potential research opportunities it
enables. In total, ABCflux includes 244 sites and 6309 monthly observations; 136 sites and 2217 monthly
observations represent tundra, and 108 sites and 4092 observations represent the boreal biome. The
database includes fluxes estimated with chamber (19 % of the monthly observations), snow diffusion (3
%) and eddy covariance (78 %) techniques. The largest number of observations were collected during the
climatological summer (June-August; 32 %), and fewer observations were available for autumn
(September-October; 25 %), winter (December-February; 18 %), and spring (March-May; 25 %).
ABCflux can be used in a wide array of empirical, remote sensing and modeling studies to improve
understanding of the regional and temporal variability in $CO_2$ fluxes, and to better estimate the terrestrial
ABZ $CO_2$ budget. ABCflux is openly and freely available online
(https://doi.org/10.3334/ORNLDAAC/1934, Virkkala et al., 2021a).








## 1. Introduction

The Arctic-Boreal Zone (ABZ), comprising the northern tundra and boreal biomes, stores approximately half the global soil organic carbon pool (Hugelius et al., 2014; Tarnocai et al., 2009; Mishra et al., 2021). As indicated by this large carbon reservoir, the ABZ has acted as a carbon sink over the past millenia due to the cold climate and slow decomposition rates (Siewert et al., 2015; Hugelius et al., 2020; Gorham, 1991). However, these carbon stocks are increasingly vulnerable to climate change, which is occurring rapidly across the ABZ (Box et al., 2019). As a result, carbon is being lost from this reservoir to the atmosphere as carbon dioxide ($CO_2$) through increased ecosystem respiration (Reco) (Schuur et al., 2015; Parker et al., 2015; Voigt et al., 2017). The impact of increased $CO_2$ emissions on global warming depends on the extent to which respiratory losses are offset by gross primary productivity (GPP), the vegetation uptake of atmospheric $CO_2$ via photosynthesis (McGuire et al., 2016; Cahoon et al., 2016).

Carbon dioxide flux measurements provide a means to monitor the net $CO_2$ balance (i.e., net ecosystem exchange; NEE, a balance between GPP and Reco) across time and space (Baldocchi, 2008; Pavelka et al., 2018). There are three main techniques used to measure fluxes at the ecosystem level that represent fluxes from plants and soils: eddy covariance, automated and manual chambers, and snow diffusion methods (hereafter diffusion; for a comparison of the techniques, see Table 1 in McGuire et al. 2012). The eddy covariance technique estimates NEE at the ecosystem scale (ca. 0.01 to 1 $km^2$ footprint) at high temporal resolution (i.e., ½ hr) using nondestructive and automated measurements (Pastorello et al., 2020). Automated and manual chamber techniques measure NEE at fine spatial scales (< 1 $m^2$) and in small-statured ecosystems, common in the tundra, where the chambers can fit over the whole plant community (Järveoja et al., 2018; López-Blanco et al., 2017). The diffusion technique can be used to measure the transport of $CO_2$ within a snowpack (Björkman et al., 2010b). The eddy covariance technique has been used globally for over three decades, and chamber and diffusion techniques for even longer.

Historically, the number and distribution of ABZ flux sites has been rather limited compared to observations in temperate regions (Baldocchi et al., 2018). Due to these data gaps, quantifying



the net annual $CO_2$ balance across the ABZ has posed a significant challenge (Natali et al., 2019;
McGuire et al., 2016; Virkkala et al., 2021b). However, over the past decade, the availability of
ABZ flux data has increased substantially. Many, but not all, of the ABZ eddy covariance sites
are a part of broader networks, such as the global FLUXNET and regional AmeriFlux, Integrated
Carbon Observation System (ICOS) and the European Fluxes Database Cluster (EuroFlux),
where data are standardized and openly available (Paris et al., 2012; Novick et al., 2018;
Pastorello et al., 2020). These networks primarily include flux and meteorological data, but do
not often include other environmental descriptions such as soil carbon stocks, dominant plant
species, or the disturbance history of a given site (but see, for example, BADM data in
Ameriflux), which are important for understanding the controls on $CO_2$ fluxes. Moreover, even
though some ABZ annual chamber measurements are included in the global soil respiration
database (SRDB) (Jian et al., 2020), and in the continuous soil respiration database (COSORE)
(Bond-Lamberty et al., 2020), standardized datasets providing ABZ $CO_2$ flux measurements
from eddy covariance, chambers, and diffusion, along with comprehensive metadata, have been
nonexistent. Such an effort would create potential for a more thorough understanding of ABZ
$CO_2$ fluxes. Therefore, compiling these flux measurements and their supporting ancillary data
into one database is clearly needed to support future modeling, remote sensing, and empirical
data mining efforts.

Arctic-Boreal $CO_2$ fluxes have been previously synthesized in a handful of regional studies
(Belshe et al., 2013; McGuire et al., 2012; Luyssaert et al., 2007; Baldocchi et al., 2018; Virkkala
et al., 2018; Natali et al., 2019; Virkkala et al., 2021b) (Fig. 1 and Table 1). One of the main
challenges in these previous efforts, in addition to the limited geographical coverage of ABZ
sites and lack of environmental descriptions, has been the variability of the synthesized seasonal
measurement periods. Most of these efforts have allowed the seasonal definitions and
measurement periods to vary across the sites, creating uncertainty in the inter-site comparison of
flux measurements. An alternative approach to define seasonality is to focus on standard time
periods such as months (Natali et al., 2019). Although focusing on monthly fluxes may result in
a small decrease in synthesizable data, because publications, particularly older ones, often
provide seasonal rather than monthly flux estimates (see e.g., (Euskirchen et al., 2012; Nykänen
et al., 2003; Björkman et al., 2010a; Oechel et al., 2000; Merbold et al., 2009)), compiling



monthly fluxes has several advantages over the seasonal fluxes. These advantages include: (i)
better comparability of measurements, (ii) ability to bypass problems related to defining seasons
across large regions, and (iii) ease of linking these fluxes to remote sensing and models.

Our goal is to build upon past synthesis efforts and compile a new database of Arctic-Boreal $CO_2$
fluxes (ABCflux version 1) that combines eddy covariance, chamber, and diffusion data at
monthly timescales with supporting environmental information to help facilitate large-scale
assessments of the ABZ carbon cycle. This paper provides a general description of the ABCflux
database by characterizing the data sources and database structure (Section 2), as well as
describing the characteristics of the database (Section 3). Additionally, we describe the main
strengths, limitations, and opportunities of this database (Section 4), and its potential utility for
future studies aiming to understand terrestrial ABZ $CO_2$ fluxes.

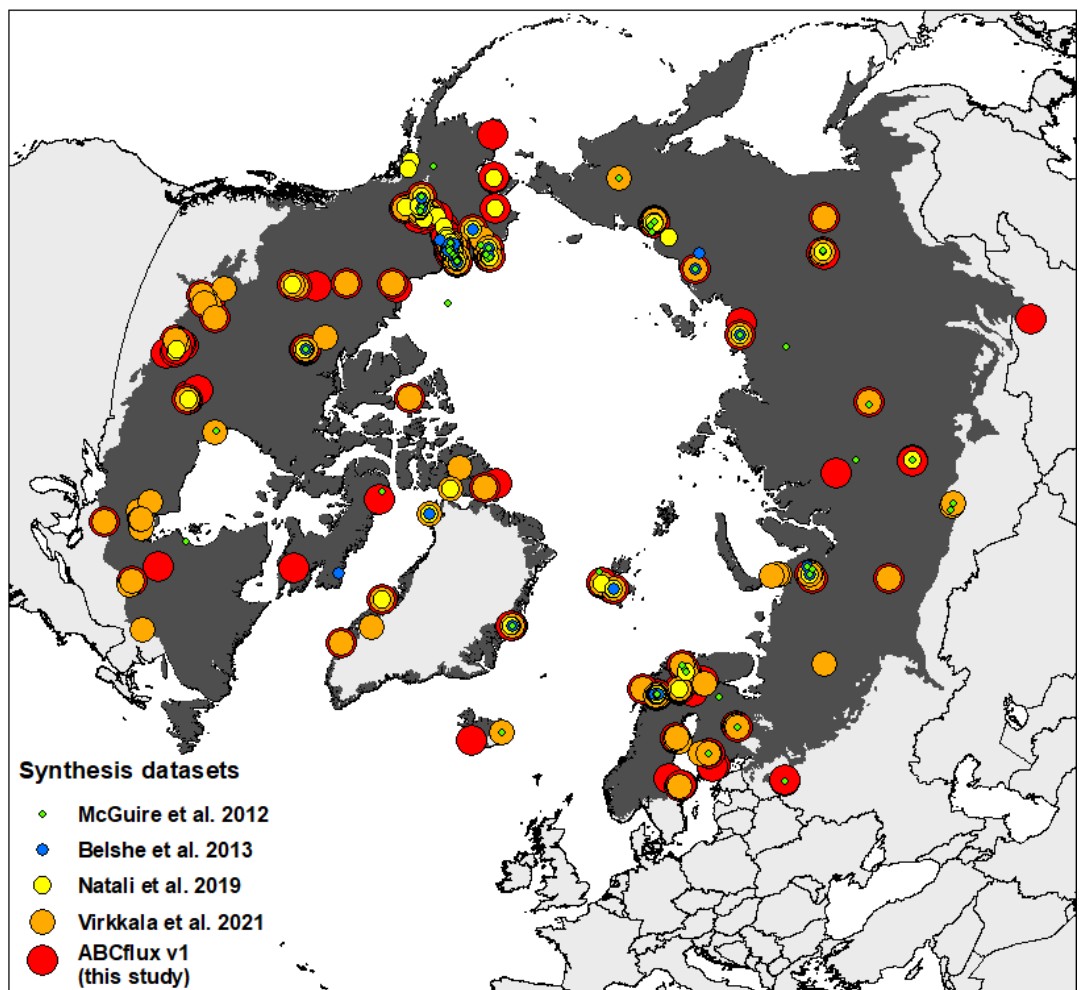


**Fig 1**. The flux site distribution in previous syntheses that focused on compiling fluxes from high latitudes (McGuire et al. 2012, Belshe et al. 2013, Natali et al. 2019, Virkkala et al. 2021b and this study (ABCflux)). The Arctic-Boreal Zone is highlighted in dark grey; countries are shown in the background. Based on the unique latitude-longitude coordinate combinations in the tundra, there were 136 tundra sites in ABCflux, 104 tundra sites in Virkkala et al. 2021b, 68 tundra sites in Natali et al., 2019, 34 tundra sites in Belshe et al. 2013, and 66 tundra sites in McGuire et al., 2012. Observations that were included in previous studies but not in ABCflux represent fluxes aggregated over seasonal, not monthly periods.




**Table 1**. A summary of past $CO_2$ flux synthesis efforts. If site numbers were not provided in the
paper, this was calculated as the number of unique sets of coordinates.

| Study | Number of sites | Synthesized fluxes and measurement techniques | Study domain | Study period | Flux aggregation |
|---|---|---|---|---|---|
| Luyssaert et al. (2007) | NA | GPP, Reco, and NEE measured with eddy covariance | Global forests (including boreal) | NA | Annual |
| McGuire et al. (2012) | 60 | GPP, Reco, and NEE measured with chambers, eddy covariance, diffusion technique and soda lime | Arctic tundra | Measurements from 1966-2009; focus on 1990-2009 | Annual, growing and winter season |
| Belshe et al. (2013) | 34 | GPP, Reco, and NEE measured with chambers, eddy covariance, diffusion technique and soda lime | Arctic tundra | Measurements from 1966-2010 | Annual, growing and winter season |
| Baldocchi et al. (2018) | 9 | GPP, Reco, and NEE measured with eddy covariance | Global (including boreal and tundra biomes) | NA (sites with 5-18 years of measurements) | Annual |
| Virkkala et al. (2018) | 117 | GPP, Reco, and NEE measured with chambers | Arctic tundra | Studies published during 2000-2016 | Growing season |
| Natali et al. (2019) | 104 | Soil respiration (or NEE) measured with chambers, eddy covariance, diffusion | Northern permafrost region | Measurements from 1989-2017, focus on 2000-2017 | Monthly or seasonal during winter |





| | | | | | |
|---|---|---|---|---|---|
| | | technique, and soda lime | | | |
| Virkkala et al. (2021b) | 148 | GPP, Reco, and NEE measured with chambers and eddy covariance | Arctic tundra and boreal biomes | 1990-2015 | Annual and growing season |
| ABCflux version 1 (this study) | 244 | GPP, Reco, and NEE (with some soil respiration and forest floor fluxes) measured with chambers, eddy covariance, and diffusion technique | Arctic tundra and boreal biomes | 1989-2020 | Monthly (whole year) |


## 2. Data and methods

ABCflux focuses on the area covered by the northern tundra and boreal biomes (>45 °N), as
characterized in (Dinerstein et al., 2017), Fig. 2)), and compiles *in-situ* measured terrestrial
ecosystem-level $CO_2$ fluxes aggregated to monthly time periods (unit: g C m$^{-2}$ month$^{-1}$).
Although the three flux measurement techniques included in ABCflux primarily measure NEE,
chamber and eddy covariance techniques can also be used to estimate GPP (the photosynthetic
flux) and Reco (comprising emissions from autotrophic and heterotrophic respiration) (Keenan
and Williams, 2018), which are also included in the database. At eddy covariance sites, GPP and
Reco are indirectly derived from NEE using partitioning methods that primarily use light and
temperature data (Lasslop et al., 2010; Reichstein et al., 2005). At chamber sites, Reco can be
measured directly with dark chambers, from which GPP can be calculated by subtracting Reco
from NEE (Shaver et al., 2007). In general, these partitioned GPP and Reco fluxes have higher
uncertainties than the NEE measurements since they are modeled based on additional data and
various assumptions (Aubinet et al., 2012). However, GPP and Reco fluxes were included in





ABCflux because these component fluxes may help to better understand and quantify the
underlying processes of land–atmosphere $CO_2$ exchange.

In addition to $CO_2$ fluxes, we gathered information describing the general site conditions (e.g.,
site name, coordinates, vegetation type, disturbance history, a categorical soil moisture variable,
and soil organic carbon stocks), micrometeorological and environmental measurements (e.g., air
and soil temperatures, precipitation, soil moisture, snow depth), and flux measurement technique
(e.g., measurement frequency, instrumentation, gap filling and partitioning method, number of
spatial replicates for chamber measurements, flux data quality), wherever possible.









**Fig 2**. Map showing the distribution and measurement technique at each site (a), and examples of
an eddy covariance tower (b), manual chamber (c), diffusion measurements (d), and two eddy
covariance towers in wetland and forest (e). Photographs were taken in Yukon-Kuskokwim
Delta, Alaska (September 2019), Kilpisjärvi, Finland (July 2016), Montmorency forest, Canada
(April 2021), and Scotty Creek, Canada (April, 2014). Image credits to: Markus Jylhä, Alex
Mavrovic, Gabriel Hould Gosselin, Chris Linder, Manuel Helbig.

2.1. Data sources
2.1.1 Literature search
We identified potential $CO_2$ flux studies and sites from prior synthesis efforts (Belshe et al.,
2013; McGuire et al., 2012; Virkkala et al., 2018; Natali et al., 2019; Virkkala et al., 2021b),
including a search of citations within and of the studies included in these prior syntheses. We
also conducted a literature search with the search words ("carbon flux" or "carbon dioxide flux"
or "NEE" or "net ecosystem exchange"), and ("arctic" or "tundra" or "boreal") in Web of
Science to ensure that our database included the most recent publications. We included studies
that reported at least NEE, presented at monthly or finer temporal resolution, and had supporting
environmental ancillary data describing the sites. We extracted our variables of interest (Section
2.3.) from these selected papers during 2018-2020. Data from line and bar plots were extracted
using Plot Digitizer (http://plotdigitizer.sourceforge.net/) and converted to our flux units (g C m$^{-2}$
month$^{-1}$) if needed. Papers including a low number of temporal replicates within a month (<3
individual measurements in summer months) and only one measurement month were
disregarded. For the spring (March-May), autumn (September-November), and winter
(December-February) months, one temporal replicate was accepted due to scarcity of
measurements outside the summer season (June-August); measurement frequency is included in
the database. Data from experimental treatments were excluded; however, we included flux data
from unmanipulated control plots. Winter chamber or diffusion measurements in forests from
Natali et al., (2019) were included in the "Ground_NEE" field, which represents forest
understory (not whole-ecosystem) NEE.



### 2.1.2. Flux repositories

We downloaded eddy covariance and supporting environmental data products from AmeriFlux
(Novick et al., 2018), Fluxnet2015 (Pastorello et al., 2020), EuroFlux database cluster (ICOS,
Carbon Extreme, Carbo Africa, GHG Europe, Carbo Italy, INGOS) (Paris et al., 2012; Valentini,
2003), and Station for Measuring Ecosystem-Atmosphere Relations (Hari et al., 2013). Data
were downloaded in 2018-2020. When only daily gap-filled data were supplied, we summed the
data to monthly time steps and recorded the percentage of gap-filled data. We did not aggregate
any repository GPP, Reco, or NEE datasets that were not gap filled. We filtered out
measurements with low turbulence conditions based on friction velocity (USTAR) thresholds
(Aubinet et al., 2012). USTAR varied among sites due to differing site-level assumptions. We
downloaded only gap filled data that met the USTAR criteria for either the tower PI or given
through the database processing pipeline. However, Fluxnet2015 provides several different
methods for determining data quality based on different USTAR criteria. In this case, we used
the Fluxnet2015 common USTAR threshold (CUT, i.e. all years at the site filtered with the same
USTAR threshold (Pastorello et al., 2020)). For observations extracted from EuroFlux, USTAR
thresholds for each site were derived as described in (Papale et al., 2006; Reichstein et al., 2005)
using night-time data. If fluxes were available for the same period both in Natali et al., (2019)
and flux repository extractions, the flux repository observations were kept in the database. Some
repositories supplied eddy covariance data version numbers, which were added to the flux
database.

### 2.1.3. Permafrost Carbon Network data solicitation

A community call was solicited in 2018 through a $CO_2$ flux synthesis workshop (Parmentier et
al., 2019, Reconciling historical and contemporary trends in terrestrial carbon exchange of the
northern permafrost-zone, 2021), whereby the network of ABZ flux researchers were contacted
and invited to contribute their most current unpublished data. This resulted in an additional 39
sites and 1372 monthly observations (see column Extraction_source).



## 2.2. Data quality screening

We screened for poor-quality data, potential unit and sign convention issues, and inaccurate coordinates. Repository data were processed and quality checked using quality flags associated with monthly data supplied by the repository processing pipeline. Fluxnet2015 and EuroFlux database include an aggregated data quality flag (fraction between 0-1, indicating percentage of measured and good-quality gap-filled data; average from daily data; 0=extensive gap-filling, 1=low gap-filling; for more details see (Pastorello et al., 2020)) which is reported for fluxes aggregated from finer temporal resolutions. We removed monthly data with a quality flag of 0. Eddy covariance data with quality flags >0 were left within the database for the user to decide on additional screening criteria. The database also includes a column describing the percentage of gap-filled data (0=no gap-filled data, 100=completely gap-filled data), however it was not used in data quality screening. These fields describe the amount and quality of the gap-filled data that need to be filled due to, for example, instrument malfunction, power shortage, extreme weather events, and periods with insufficient turbulence conditions.

We further screened for spatial coordinate accuracy by visualizing the sites on a map. If a given site was located in water or had imprecise coordinates, the site researchers were contacted for more precise coordinates. We screened for potential duplicate sites and observations that were extracted from different data sources. Duplicate NEE extracted from papers that were also extracted from flux repositories were compared to estimate uncertainties associated with using Plot Digitizer as a means for extracting monthly fluxes. A linear regression between paper (Plot Digitizer) and repository extraction showed that data extracted using Plot Digitizer were highly correlated with data from online databases, providing confidence in estimates extracted using Plot Digitizer ($R^2$=0.91, slope = 1.002, n=192). Out of these duplicate observations, we only kept the data extracted from the repository in the database. Finally, we asked site principal investigators (PIs) to verify that the resulting information was correct.

## 2.3. Database structure and columns

The resulting ABCflux database includes 94 variables: 16 are flux measurements and associated metadata (e.g., NEE, measurement date and duration), 21 describe flux measurement methods





(e.g., measurement frequency, gap-filling method), 49 describe site conditions (e.g., soil
moisture, air temperature, vegetation type), and 8 describe the extraction source (e.g., primary
author or site PI, citation, data maturity). 61 variables are considered static and thus do not vary
with repeated measurements at a site (e.g., site name, coordinates, vegetation type), while 33
variables are considered dynamic and vary monthly (e.g., soil temperature). Table 2 includes a
description of each of the 94 variables, as well as the proportion of monthly observations present
in each column. ABCflux is shared as a comma separated values (csv) file with 6309 rows;
however, not all the rows have data in each column (indicated by NA).

We refer to all fields included in ABCflux as observations although we acknowledge that, for
example, GPP and Reco are indirectly derived variables at eddy covariance sites, and that some
flux and ancillary data can also be partly gap-filled. Further, our database does not include the
actual raw observations, rather it provides monthly aggregates. Positive values for NEE indicate
net $CO_2$ loss to the atmosphere (i.e., $CO_2$ source) and negative numbers indicate net $CO_2$ uptake
by the ecosystem (i.e., $CO_2$ sink). For consistency, GPP is presented as negative (uptake) values
and Reco as positive.

**Table 2**. Database variables and the proportion of monthly observations in each variable. There
are in total 6309 monthly observations in the database.

| Variable | Variable description and units | Details | Proportion of monthly observations having data |
|---|---|---|---|
| id | ID given to each individual monthly entry at each site | | 100% |
| Study_ID | ID given to study/site entry (see Details) | (PI/first author of publication)_(site name)_(tower/chamber)_(#); Eg., Schuur_EML_Tower_1. Note that there might be several chamber (or tower) Study_IDs for one site. | 100% |



| Study_ID_Short | ID given to study/site entry (see Details), individual chamber plots within a site not differentiated | (PI/first author of publication)_(site name)_(tower/chamber)_(#); Eg., Schuur_EML_Tower_1. | 100% |
|---|---|---|---|
| Site_Name | Site name as specified in data source | Usually the location name | 100% |
| Site_Reference | A more specific name used in data source | For towers, this is often the acronym for the site, and for chambers, this is the name of the particular chamber plot | 95% |
| Data_contributor_or_Author | Data contributor(s) or primary author(s) associated with data set or publication | If you use unpublished data or data from flux repositories (see Extraction_source), please contact this person | 100% |
| Latitude | Decimal degrees, as precise as possible | | 100% |
| Longitude | Decimal degrees, as precise as possible | Negative longitudes are west from Greenwich | 100% |
| Email | Primary author email | | 93% |
| ORCID | personal digital identifier: https://orcid.org/ | | 29% |
| Citation | Journal article, data citation, and/or other source (online repository, PI submitted, etc.). | | 70% |
| Data_adder | The person(s) who added the data to the database | Primarily researchers working at Woodwell | 100% |
| Data_availability | Current availability of data: data available in a published paper, in an open online data repository, in an already published synthesis, or user contributed | Published_Paper, Published_Online, Published_Synthesis, User_Contributed | 100% |
| Data_maturity | Current maturity of data | Preliminary, Processed, Published, Reprocessed | 100% |



| Extraction_source | Data source | paper, Virkkala or Natali syntheses, Euroflux, Fluxnet 2015, PI, Ameriflux, SMEAR, ORNL DAAC, Pangaea | 100% |
|---|---|---|---|
| Biome | Biome of the site | Boreal, Tundra | 100% |
| Veg_type | A detailed vegetation type for the site | B1=cryptogram, herb barren; B2=cryptogram barren complex; B3=noncarbonate mountain compled; B4=carbonatemountain complex; G1=rush/grass, forb, cryptogram tundra; G2=graminoid, prostrate dwarf-shrub, forb tundra; G3=nontussock sedge, dwarf-shrub, moss tundra; G4=tussock-sedge, dwarf-shrub, herb tundra; P1=prostrate dwarf-shrub, herb tundra; P2=prostrate/hemiprostrate dwarf-shrub tundra; S1=erect dwarf-shrub tundra; S2=low-shrub tundra; W1=sedge/grass, moss wetland; W2=sedge, moss, dwarf-shrub wetland; W3=sedge, moss, low-shrub wetland; DB=deciduous broadleaf forest; EN=evergreen needleleaf forest; DN=deciduous needleleaf forest; MF=mixed forest; SB=sparse boreal vegetation; BW=boreal wetland or peatland, following Watts et al. (2019). For more details about the tundra vegetation types, see Walker et al. (2005). These classes were classified based on information in Site_Reference and Veg_detail columns, or were contributed by the site PI. | 100% |
| Veg_type_Short | A more general vegetation type for the site | B=barren tundra; G=graminoid tundra; P=prostrate dwarf-shrub tundra; S=shrub tundra; W=tundra wetland; DB=deciduous broadleaf forest; EN=evergreen needleleaf forest; DN=deciduous needleleaf forest; MF=mixed forest; SB=sparse boreal vegetation; BW=boreal wetland or peatland. For more details about the tundra vegetation types, see Walker et al. (2005). These classes were classified based on information in Site_Reference and Veg_detail columns, or were contributed by the site PI. | 100% |
| Veg_detail | Detailed vegetation description from data source/contributor | | 96% |
| Country | Country of the site | | 100% |
| Permafrost | Reported presence or absence of permafrost | Yes, No | 72% |



| | | | |
|---|---|---|---|
| Disturbance | Last disturbance | Fire, Harvest, Thermokarst, Drainage, Grazing, Larval Outbreak, Drought | 30% |
| Disturb_year | Year of last disturbance | Numeric variable, 0 = annual (e.g., annual grazing) | 23% |
| Disturb_severity | Relative severity of disturbance | High, Low | 11% |
| Soil_moisture_class | General descriptor of site moisture | Wet = At least sometimes inundated or water table close to surface. Dry = well-drained. | 56% |
| Site_activity | Describes whether the site is currently active (i.e., measurements conducted each year) | Yes, No. Eddy covariance information was extracted from https://cosima.nceas.ucsb.edu/carbon-flux-sites/ by assuming that sites that were active in 2017 are still continuing to be active. We used our expertise to define active chamber sites that have measurements at least during each growing season. | 60% |
| Meas_year | Year in which data were recorded | | 100% |
| Season | Season in which data were recorded | summer, autumn, winter, spring (based on climatological seasons) | 100% |
| Interval | Measurement month | | 100% |
| Start_day | Start day of the measurement | | 100% |
| End_day | End day of the measurement | | 100% |
| Duration | Number of days during the measurement month | Should be the same as End_Day because this database compiles monthly fluxes | 100% |
| Start_date | Date on which measurement starts | dd/mm/yyyy | 100% |
| End_date | Date on which measurement ends | dd/mm/yyyy | 100% |
| NEE_gC_m2 | Net Ecosystem Exchange (g C-$CO_2$ m$^{-2}$ for the entire measurement interval) | Convention: -ve is uptake, +ve is loss. | 91% |





| | | | |
|---|---|---|---|
| GPP_gC_m2 | Gross Primary Productivity (g C-CO$_2$ m$^{-2}$ for the entire measurement interval) | Report as -ve flux | 68% |
| Reco_gC_m2 | Ecosystem Respiration (g C-CO$_2$ m$^{-2}$ for the entire measurement interval) | Report as +ve flux | 73% |
| Ground_NEE_gC_m2 | Forest floor Net Ecosystem Exchange, measured with chambers (g C-CO2 m-2 for the entire measurement interval) | Convention: -ve is uptake, +ve is loss. Chamber measurements from (primarily rather treeless) wetlands are included in the NEE_gC_m2 column. | 4% |
| Ground_GPP_gC_m2 | Forest floor Ecosystem Respiration, measured with chambers (g C-CO2 m-2 for the entire measurement interval) | Report as -ve flux. Chamber measurements from (primarily rather treeless) wetlands are included in the GPP_gC_m2 column. | 1% |
| Ground_Reco_gC_m2 | Forest floor Gross Primary Productivity, measured with chambers (g C-CO$_2$ m$^{-2}$ for the entire measurement interval) | Report as +ve flux. Chamber measurements from (primarily rather treeless) wetlands are included in the Reco_gC_m2 column. | 2% |
| Rsoil_gC_m2 | Soil respiration, measured with chambers (g C-CO$_2$ m$^{-2}$ for the entire measurement interval) | Report as +ve flux | 4% |
| Flux_method | How flux values were measured | EC=eddy covariance, Ch=chamber, Diff=diffusion methods. No observations from experimental manipulation plots | 100% |
| Flux_method_detail | Details related to how flux values were measured: closed- and open-path eddy covariance, mostly manual chamber measurements, mostly automated chamber measurements, a combination of chamber and cuvette measurements, diffusion measurements through the snowpack, chamber measurements on top of snow | EC_closed, EC_open, EC_enclosed, EC_open & closed, EC_enclosed, Chambers_mostly_manual, Chambers_mostly_automatic, Chambers_CUV, Snow_diffusion, Chambers_snow, NA | 93% |



| Measurement_frequency | Frequency of flux measurements | >100 characterizes high-frequency measurements. Manual chamber and diffusion techniques often have values between 1 and 30; 1=measurements done during one day of the month, 30=measurements done daily throughout the month. | 100% |
|---|---|---|---|
| Diurnal_coverage | Times of day covered by flux measurements | Day, Day and Night | 90% |
| Partition_method | Method used to partition NEE into GPP and Reco | Reichstein (night time=Reco partitioning), Lasslop (bulk/day-time partitioning), Reco_measured, ANN, or GPP=Reco-NEE (for chamber sites) | 16% |
| Spatial_reps_chamber | Number of spatial replicates for the chamber plot | Usually, but not always, several chamber plots are measured to assure the representativeness of measurements | 71% |
| Gap_fill | Gap filling method | e.g., Average, Linear interpolation, Lookup table, MDS (marginal distribution sampling), Light/temperature response, Neural network, a combination of these, or a longer description related to chamber measurements | 70% |
| Gap_perc | % of NEE data that was gap-filled in the measurement interval (relative to standard measurement time step) | | 17% |
| Tower_QA.QC.NEE.flag | Overall monthly quality flag for eddy covariance aggregated observations; fraction between 0-1, indicating percentage of measured and good-quality gap-filled data | 0=extensive gap-filling, 1=low gap-filling | 44% |
| QA.QC.source | The source for the overall quality information for the eddy covariance observations | 0=Fluxnet2015, 1=Euroflux | 37% |
| Precip_int_mm | Total precipitation during measurement interval (mm) | | 37% |
| Tair_int_C | Mean air temperature during measurement interval (°C) | | 72% |



| | | | |
|---|---|---|---|
| Tsoil_C | Mean soil Temperature during measurement interval (°C) | | 74% |
| Soil_moisture_perc | Mean soil moisture during the measurement interval (% by volume) | | 35% |
| Thaw_depth_cm | Mean thaw depth during the measurement interval (cm) | Report with positive values | 6% |
| Tsoil_depth_cm | Depth of soil temperature measurement below surface (cm) | | 46% |
| Moisture_depth_cm | Depth of soil moisture measurement below surface (cm) | | 31% |
| ALT_cm | Active layer thickness (cm; maximum thaw depth), will change annually | Report with positive values | 15% |
| WTD_cm | Mean water table depth during the measurement interval (cm); Positive is below the surface, negative is above (inundated) | | 7% |
| Snow_depth_cm | Mean snow depth during the measurement interval (cm) | | 14% |
| VPD_Pa | Mean vapour pressure deficit during the measurement interval (Pa) | | 30% |
| ET_mm | Total evapotranspiration during the measurement interval (mm) | | 4% |
| PAR_W_m2 | Mean photosynthetically active radiation during measurement interval (W m$^{-2}$) | | 5% |





| | | | |
|---|---|---|---|
| PAR_PPFD_umol_m2_s | Mean photosynthetically active radiation during measurement interval (measured in Photosynthetic Photon Flux Density, PPFD; micromol m-2 s-1) | | 11% |
| Precip_ann_mm | Mean annual precipitation (mm), from site or nearby weather station as a general site descriptor. This should describe the longer-term climate for the site rather than a few years of study. | | 80% |
| Tair_ann_C | Mean annual air temperature (°C), from site or nearby weather station as a general site descriptor. This should describe the longer-term climate for the site rather than a few years of study. | | 79% |
| Met_source | Data source and years used to calculate mean annual temperature/precipitation | | 50% |
| Elevation_m | Elevation above sea level (m) | | 65% |
| LAI | Leaf Area Index | | 23% |
| SOL_depth_cm | Soil organic layer depth (cm) | | 23% |
| perc_C | Soil carbon percentage (%) | | 7% |
| perc_C_depth_cm | Depth at which soil carbon % was measured (cm) | | 7% |
| C_dens_kgC_m2 | Soil carbon per unit area (kg C m⁻²) | | 16% |
| C_dens_depth_cm | Depth to which Soil organic carbon per unit area was | | 8% |





| | estimated (cm) | | |
|---|---|---|---|
| AGB_kgC_m2 | Above ground biomass (kg C m$^{-2}$) | | 11% |
| AGB_type | Types of above ground vegetation included in the AGB measurement | Trees, shrubs, graminoids, mosses, lichens | 13% |
| Soil_type | General soil type, including source (e.g., USDA, CSSC, NCSCD) | | 42% |
| Soil_type_detail | Detailed soil type description, if available | | 9% |
| Citation_Data_Overlap | Another citation for the site | | 13% |
| High_freq_availability | Availability of high-frequency data | | 17% |
| Light_response_method_chamber | Details related to how the varying light response conditions were considered in chamber measurements | | 5% |
| PAR_cutoff_umol_m2_second | PAR level used to define night-time data and apply partitioning method (umol PAR m$^{-2}$ second$^{-1}$) | | 17% |
| Aggregation_method | Method used to aggregate data to measurement interval | | 58% |
| Instrumentation | Description of instrumentation used | | 68% |
| Tower_Version | Version number of the eddy covariance dataset from the extraction source | | 21% |
| Spatial_variation_technique | Technique used to quantify spatial variation for flux measurements | e.g., standard error of replicate measurements for chambers, spatial error based on footprint partitioning for towers | 10% |



| Method_error_NEE_gC_m2 | RMSE or other bootstrapped error of model fit for NEE (g C-$CO_2$ m$^{-2}$ for the entire measurement interval) | | 23% |
|---|---|---|---|
| Method_error_technique | Technique used to quantify method errors for flux measurements | e.g., gap-filling and partitioning errors or uncertainty in data-model fit: bootstrap, MCMC, RMSE fit, etc. | 1% |
| Tower_Data_restriction | | | 12% |
| Tower_Corrections | Details related to processing corrections employed, including time, duration, and thresholds for u* and heat corrections | | 32% |
| Other_data | Other types of data from the data source that may be relevant | | 7% |
| Notes_SiteInfo | Any other relevant information | | 20% |
| Notes_TimeVariant | Any other relevant information | | 59% |




2.4. Database visualization
The visualizations in this paper were made with the full ABCflux database using each site-month
as a unique data point (from now on, these are referred to as monthly observations) and the sites
listed in the "Study_ID_Short" field. We visualized these across the vegetation types
("Veg_type_Short"), countries ("Country"), biomes ("Biome"), and measurement method
("Flux_method").

To understand the distribution and representativeness of monthly observations and sites across
the ABCflux as well as the entire ABZ, we used geospatial data to calculate the aerial coverages
of each vegetation type and country. Vegetation type was derived from the European Space
Agency Climate Change Initiative's (ESA CCI) land cover product aggregated and resampled to



0.0083° for the boreal biome (Lamarche et al., 2013) and the raster version of the Circumpolar
Arctic Vegetation Map (CAVM) for the tundra biome resampled to the same resolution as the
ESA CCI product (Raynolds et al., 2019). ESA CCI layers were reclassified by grouping land
cover types to the same vegetation type classes represented by ABCflux: boreal wetland and
peatland (from now on, boreal wetland; classes 160, 170, 180 in ESA CCI product), deciduous
broadleaf forest (60-62), evergreen needleleaf forest (70-72), deciduous needleleaf forest (80-
82), mixed forest (90), and sparse and mosaic boreal vegetation (40, 100, 100, 120, 121, 122,
130, 140, 150, 151, 152, 153, 200, 201, 202). Croplands (10, 11, 12, 20, 30) and urban areas
(190) were removed. We used the five main physiognomic classes from CAVM in the tundra.
Glaciers and permanent water bodies included in either of these products were removed. Note
that in ABCflux and for the site-level visualizations in this paper, vegetation type for each of the
flux sites was derived from site-level information, not these geospatial layers. These same
glacier, water, and cropland masks were applied to the country boundaries (Natural Earth - Free
vector and raster map data at 1:10m, 1:50m, and 1:110m scales, 2021) to calculate the terrestrial
area of each country.

## 3. Database summary
### 3.1. General characteristics of the database
ABCflux includes 244 sites and 6309 monthly observations, out of which 136 sites and 2217
monthly observations are located in the tundra (54 % of sites and 52 % of observations from
North America, 46 % and 48 % from Eurasia), while 108 sites and 4092 monthly observations
are located in the boreal biome (59 % of sites and 58 % of observations from North America, 41
% and 42 % from Eurasia) (Table 3). The largest source of flux data are the flux repositories (48
% of the monthly observations), while flux data extracted from papers or contributed by site PIs
amount to 30 % and 22 % of the monthly observations, respectively. The database primarily
includes sites in unmanaged ecosystems, but it does contain a small number (6) of sites in
managed forests.





**Table 3**. General statistics of the database. Number of monthly $CO_2$ flux measurements and sites derived from eddy covariance, chamber, and diffusion techniques, and the proportion of data coming from different data sources. Note that some of the data extracted from flux repositories and papers were further edited by the PIs; this information can be found in the database. For this table, observations that were fully contributed by the PI were considered as PI-contributed.

| Flux measurement technique | Number of sites | Number of observations | Number of observations derived using different eddy covariance and chamber techniques | Number of observations extracted from different data sources |
|---|---|---|---|---|
| Eddy covariance | Total: 119<br>Tundra: 47<br>Boreal: 72 | Total: 4957<br>Tundra: 1406<br>Boreal: 3551 | Open-path: 1988<br>Closed path: 2085<br>Both: 245<br>Enclosed: 240<br>No information available: 399 | Flux repository: 2775<br>Published: 810<br>PI-contributed: 1350 |
| Chamber | Total: 104<br>Tundra: 73<br>Boreal: 31 | Total: 1166<br>Tundra: 708<br>Boreal: 458 | Manual: 435<br>Automated: 696<br>No information available: 35 | Flux repository: 243<br>Published: 901<br>PI-contributed: 22 |
| Diffusion | Total: 21<br>Tundra: 16<br>Boreal: 5 | Total: 186<br>Tundra: 103<br>Boreal: 83 | | Flux repository: 0<br>Published: 186<br>PI-contributed: 0 |

The majority of observations in ABCflux have been measured with the eddy covariance technique (119 sites and 4957 monthly observations), whereas chambers and diffusion methods were used at 125 sites and 1352 observations (Table 3). About 46 % of the eddy covariance measurements are based on gas analyzers using closed-path technology (including enclosed analyzers), 40 % are based on open-path technology, 5 % include both and 8 % are unknown. 52 % of chamber measurements were automated chambers (monitoring the fluxes continuously throughout the growing season). Only 3 % of the measurements were completed using diffusion methods during the winter. Chamber and diffusion studies were primarily from the tundra and the sparsely treed boreal wetlands, but a few studies with ground surface $CO_2$ fluxes from forests





(i.e., capturing the ground cover vegetation and not the whole ecosystem) are also included in
their own fields so that they can not be mixed up with ecosystem-scale measurements
("Ground_NEE_gC_m2", "Ground_GPP_gC_m2", "Ground_Reco_gC_m2"). Further, a few soil
$CO_2$ flux sites measuring fluxes primarily on unvegetated surfaces during the non-growing
season are included in the database ("Rsoil_gC_m2"). These were included in the database
because ground surface or soil fluxes during the non-growing season can be of similar magnitude
to the ecosystem-level fluxes when trees remain dormant (Ryan et al., 1997; Hermle et al., 2010).
Therefore, these ground or soil fluxes could potentially be used to represent ecosystem-level
fluxes during some of the non-growing season months. However, we did not make an extensive
literature search for these observations, rather we compiled observations if they came up in our
NEE search. Therefore, the data in these ground surface and soil flux columns represents only a
portion of such available data across the ABZ.

The geographical coverage of the flux data is highly variable across the ABZ, with most of the
sites and monthly observations coming from Alaska (37 % of the sites and 28 % of the monthly
observations), Canada (19 % and 29 %), Finland (7 % and 15 %), and Russia (14 % and 13 %)
(Fig. 3). The sites cover a broad range of vegetation types, but were most frequently measured in
evergreen needleleaf forests (23 % of the sites and 37 % of the monthly observations) and
wetlands in the tundra or boreal zone (30 % and 27 %) (Fig. 4). The northernmost and
southernmost ecosystems had fewer sites and observations than more central ecosystems (barren
tundra: 45% of the sites and 3 % of the monthly observations, prostrate shrub: 2 % and <1 %,
deciduous broadleaf forest: 1 % and 3 %, deciduous needleleaf forest: 5 % and 4 %, mixed forest
<1 % and <1 %). ABCflux includes sites experiencing various types of disturbances, with the
majority of disturbed sites encountering fires (24 sites and 901 monthly observations),
thermokarst (4 sites and 113 monthly observations), or harvesting (6 sites and 258 monthly
observations). However, ABCflux is dominated by sites in relatively undisturbed environments
or sites lacking disturbance information (only 20 % of the sites and 30 % of the monthly
observations include disturbance information).



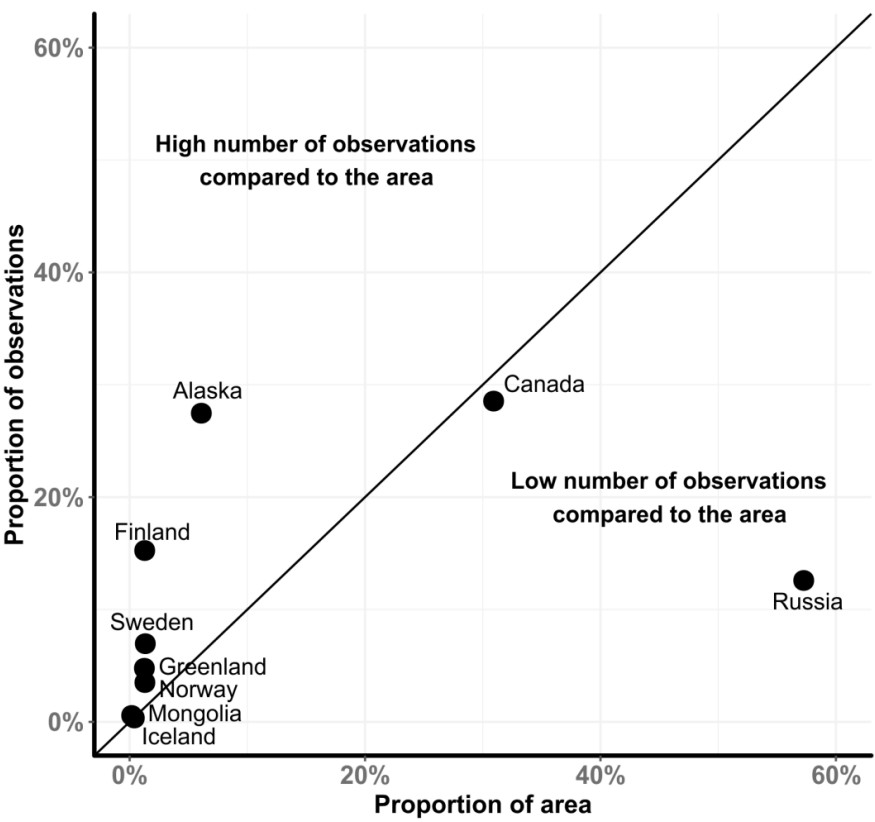


**Fig 3**. The proportion of monthly observations in each country/region compared to the

proportion of the areal extent of the country/region across the entire Arctic-Boreal Zone. Ideally,

points would be close to the 1:1 line (i.e., large countries/regions have more observations than

small countries/regions). Permanent water bodies, croplands, and urban areas were masked from

the areal extent calculation.









**Fig 4**. The proportion of monthly observations in each vegetation type colored by the flux
measurement technique (a) and the proportion of the areal extent of each vegetation type across
the entire Arctic-Boreal Zone (b). Permanent water bodies, croplands, and urban areas were
masked from the areal extent calculation. Sparse boreal vegetation class in the vegetation map
includes vegetation mixtures and mosaics.

ABCflux spans a total of 31 years (1989-2020), but the largest number of monthly observations
originate from 2000-2015 (80 % of the data) (Fig. 5). The largest number of measurements were
conducted during the summer (June-August; 32 %) and the least during the winter (November-
February; 18 %) (Fig. 5). The overall eddy covariance data quality and gap-filled data percentage
were lowest during the winter compared to other seasons (0.76 compared to 0.8-0.85 for overall
data quality, 0=extensive gap-filling, 1=low gap-filling; 69 % compared to 47 to 59 % for gap-
filled data percentage).

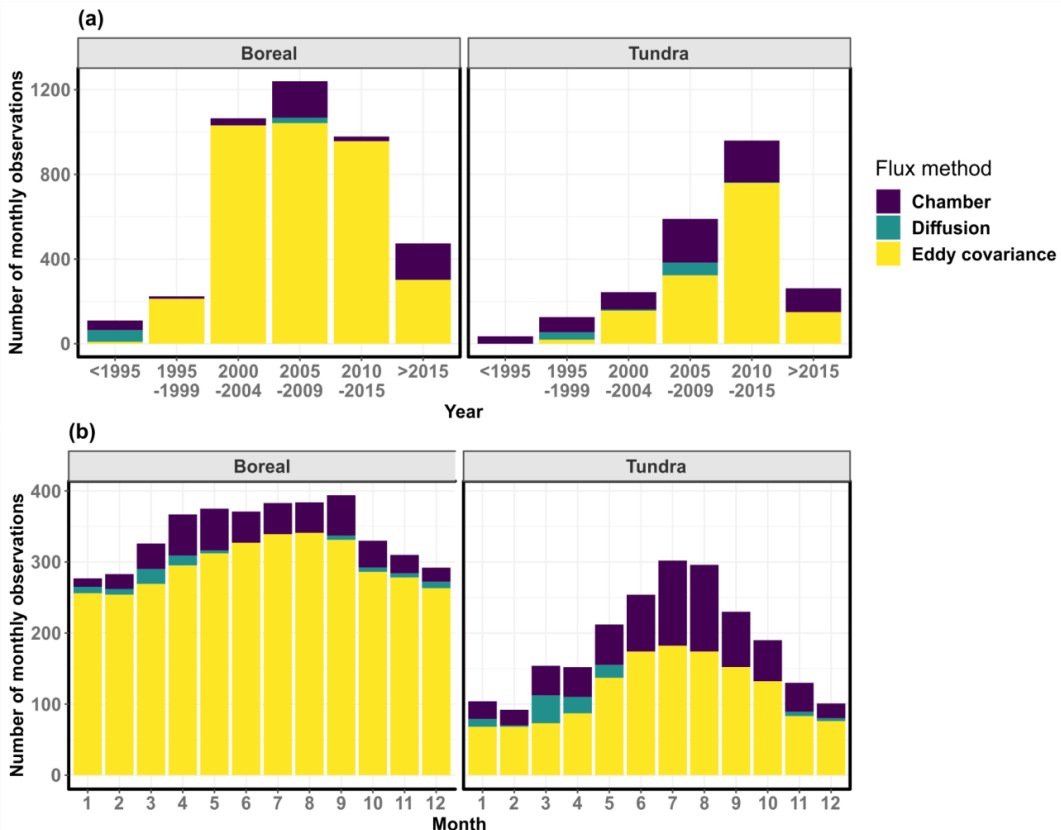


**Fig 5**. Histograms showing the number of monthly measurements across five-year periods (a)

and across the tundra and boreal biomes (b). The bar plots are colored by the flux measurement

technique. Chambers in the boreal biome measured fluxes in treeless or sparsely treed areas

(primarily wetlands).


3.2. Coverage of ancillary data

All of the observations in ABCflux include information describing the site name, location,
vegetation type, NEE, measurement technique (eddy covariance/chamber/diffusion), and how the
data were compiled (Table 2). Details about the measurement technique (e.g., open or closed-
path eddy covariance, manual or automated chambers) are included in 93 % of sites and 93 % of
monthly observations. Most of the monthly observations further include information about





permafrost extent ( 67 % of the sites and 72 % of the monthly observations), or soil moisture
state (47 % of the sites and 56 % of the monthly observations). Data describing air temperature,
soil temperature, precipitation, and soil moisture are included in 71, 73, 37, and 35 % of monthly
observations, respectively. Some ancillary variables have low data coverage, such as soil organic
carbon stocks (16 % of the monthly observations) or active layer thickness (15 % of the monthly
observations).

3.3. Coverage and distribution of flux data

There are 110 sites and 4290 monthly observations for GPP, 121 sites and 4603 monthly
observations for Reco, and 212 sites and 5759 monthly observations for NEE in ABCflux.
Monthly values range from -2 to -516 g C m$^{-2}$ month$^{-1}$ for GPP, from 0 to 550 g C m$^{-2}$ month$^{-1}$
for Reco, and from -376 to 95 g C m$^{-2}$ month$^{-1}$ for NEE (Table 4). NEE is typically negative
during the summer (i.e., net $CO_2$ sink) and mostly positive during other seasons (i.e., net $CO_2$
source) (Fig. 6). Out of all site and year combinations, annual cumulative NEE (the sum of
monthly NEE values for each year and site) can be calculated for 267 site-years. An average
annual NEE calculated based on the site-level averages from 1995 to 2020 is -27.9 g C m$^{-2}$ year$^{-1}$
(SD 85.4) for the entire region, -35.5 g C m$^{-2}$ year$^{-1}$ (SD 93.7) for the boreal biome, and -3.3 g C
m$^{-2}$ year$^{-1}$ (SD 44.2) for the tundra. However, these averages do not account for the spatial or
temporal distribution of the observations, and therefore represent coarse summaries of the
database.






**Fig 6**. The distribution of net ecosystem exchange (NEE; a), gross primary productivity (GPP;

b), and ecosystem respiration (Reco; c) across the months, colored by the flux measurement

technique. Positive numbers for NEE indicate net $CO_2$ loss to the atmosphere (i.e., $CO_2$ source)

and negative numbers indicate net CO2 uptake by the ecosystem (i.e., $CO_2$ sink). For





consistency, GPP is presented as negative values and Reco as positive. The boxes correspond to
the 25th and 75th percentiles. The lines denote the 1.5 IQR of the lower and higher quartile,
where IQR is the inter-quartile range, or distance between the first and third quartiles. There is
not much chamber data from the boreal regions as they capture NEE only at treeless wetlands.

**Table 4**. Mean and standard deviation of monthly observations of net ecosystem exchange
(NEE), gross primary productivity (GPP), and ecosystem respiration (Reco) in g C m$^{-2}$ month$^{-1}$.
Seasons were defined based on the climatological definition (autumn: September-November;
winter: December-February; spring: March-May; summer: June-August). Positive numbers for
NEE indicate net $CO_2$ loss to the atmosphere (i.e., $CO_2$ source) and negative numbers indicate
net $CO_2$ uptake by the ecosystem (i.e., $CO_2$ sink). For consistency, GPP is presented as negative
values and Reco as positive.

| Biome | Climatological season | Mean monthly NEE | Mean monthly GPP | Mean monthly Reco | Standard deviation of monthly NEE | Standard deviation of monthly GPP | Standard deviation of monthly Reco |
|---|---|---|---|---|---|---|---|
| Boreal | spring | -5 | -40 | 34 | 25 | 49 | 32 |
| Boreal | summer | -35 | -163 | 124 | 36 | 79 | 71 |
| Boreal | autumn | 14 | -38 | 52 | 18 | 45 | 46 |
| Boreal | winter | 11 | -3 | 14 | 8 | 19 | 20 |
| Tundra | spring | 6 | -11 | 18 | 9 | 16 | 14 |
| Tundra | summer | -26 | -72 | 48 | 38 | 60 | 30 |



| Tundra | autumn | 10 | -14 | 21 | 21 | 30 | 15 |
|--------|--------|----|-----|----|----|----|----|
| Tundra | winter | 9  | -2  | 12 | 10 | 9  | 11 |



## 4. Strengths, limitations, and opportunities

ABCflux provides several opportunities for an improved understanding of the ABZ carbon cycle. It can be used to calculate both short- and longer-term monthly, seasonal, or annual flux summaries for different regions, or it can be combined with remote sensing and other gridded data sets to build monthly statistical and process-based models for $CO_2$ flux upscaling. ABCflux can further be utilized to study the inter- and intra-annual $CO_2$ flux variability resulting from climate and environmental change. The site distribution in ABCflux can also be used to evaluate the extent of the current flux network and identify under-sampled regions. From a methodological perspective, data users can compare fluxes estimated with the different measurement techniques which can help understand the uncertainties associated with individual techniques. However, there are also some uncertainties that the data user should be aware of when using ABCflux, which we describe below.

### 4.1. Comparing fluxes estimated with different techniques

The ABCflux database comprises aggregated observations using eddy covariance, chamber, and diffusion methods. These methods measure $CO_2$ fluxes at different spatiotemporal resolutions and are based on different assumptions. The eddy covariance technique is currently the primary method to monitor long-term trends in ecosystem $CO_2$ fluxes (Baldocchi et al., 2018; Baldocchi, 2008), and the majority of observations in ABCflux (79 %) have been made using the technique. Transforming high-frequency eddy covariance measurements to budgets includes several processing steps that can, without harmonization and standardization of these steps (Baldocchi et al., 2001; Pastorello et al., 2020), lead to highly different budget estimates (Soloway et al., 2017). It is also important to acknowledge that the extent and size of the tower footprint differs across the sites due to differences in the height of the tower and the direction and magnitude of



the wind (Chu et al., 2021). When fluxes are aggregated over longer time periods to cumulative
budgets, one generally assumes the tower footprint remains relatively constant, capturing fluxes
from a similar part of the ecosystem (i.e., the assumption that monthly observations within one
site in ABCflux can be reliably compared with each other); but note that at shorter time periods
this might not be the case (Pirk et al., 2017; Chu et al., 2021).

The different gas analyzer technologies also play an important role for the fluxes estimated with
the eddy covariance technique. Sites located in the most northern and remote parts of the ABZ
experience a drop in irradiation during autumn and winter which limits solar power availability
for eddy covariance measurements. Closed-path systems require more power to run than open-
path sensors, but open-path sensors are known to have larger uncertainties. For example, open-
path eddy covariance sensors have been shown to incorrectly estimate NEE due to the self-
heating effect of the analyzer, which can result in systematically higher net $CO_2$ uptake
compared to closed-path sensors (Kittler et al., 2017a); however, this pattern was not clearly
observed in ABCflux when across-site comparisons were made. Furthermore, wintertime fluxes
indicating $CO_2$ uptake can be erroneous due to the limited ability of the gas analyzer to resolve
very high frequency turbulent eddies (Jentzsch et al., 2021). Recently, some types of open-path
infrared gas analysers have been found to be prone to biases in NEE that scale with sensible heat
fluxes in all seasons rather than with self-heating (Wang et al., 2017; Helbig et al., 2016).

While using eddy covariance to estimate small-scale spatial variability in NEE is challenging
(McGuire et al., 2012), this can be accomplished with chamber and diffusion techniques.
Chamber measurements can be done in highly heterogeneous environments as long as chamber
closure can be guaranteed; however, most of the chamber measurements in ABCflux have been
conducted in relatively flat and homogeneous graminoid- and wetland-dominated vegetation
types. Most chamber sites in ABCflux include ca.10-20 individual plots in total from ca. 3-5 land
cover types where fluxes are being measured (Virkkala et al., 2018). Chambers can also provide
more direct estimates of Reco and GPP relative to eddy covariance-derived fluxes, and are
therefore useful for estimating the magnitude and range of those component fluxes. However,
manual chamber and diffusion measurements are laborious and have limited temporal
representation, particularly during the non-growing season when they often have only one





monthly temporal replicate in ABCflux (McGuire et al., 2012; Fox et al., 2008). Automated
chamber measurements during the non-growing season are also rare in ABCflux.

Because of these methodological differences across the eddy covariance, chamber and diffusion
techniques, comparing fluxes between the methods may result in inconsistencies (Fig. 6). It has
been shown that chamber measurements can be both larger or smaller than the fluxes estimated
with eddy covariance (Phillips et al., 2017). This difference can be related to the uncertainties
with the eddy covariance technique as described above, or to the uncertainties with the chamber
technique (e.g., accurate determination of chamber volume, pressure perturbations, temperature
increase during the measurement, collars disturbing the ground and causing plant root excision).
The differences can also be due to the mismatch between the chamber and tower footprints (<1
m vs. 250–3000 m radii over the measurement equipment, respectively) and the difficulty of
extrapolating local chamber measurements to landscape scales (Marushchak et al., 2013; Fox et
al., 2008). However, several studies have also shown good agreement across the eddy covariance
and chamber measurements (Laine et al., 2006; Wang et al., 2013; Eckhardt et al., 2019; Riutta
et al., 2007). Potential mismatches may also be due to a bias towards daytime measurements in
manual chamber measurements (see field "Diurnal_coverage"). During daytime, plants are
actively photosynthesizing whereas respiration is the dominant flux at night (López-Blanco et al.,
2017). Presumably because of these day vs. night-time differences, we observed stronger sink
strength in manual chamber measurements compared to other flux measurements in ABCflux,
even though eddy covariance measurements have also been observed to underestimate night-time
$CO_2$ loss. This underestimation in night-time eddy covariance measurements is due to suppressed
turbulent exchange linked to stable atmospheric stratification, and systematic biases due to
horizontal advection (Aubinet et al., 2012). Despite these uncertainties, including fluxes
estimated with all of these techniques into one database improves the understanding of
underlying variability of landscape-scale flux estimates. Indeed, there are roughly 10 sites in
ABCflux that include both eddy covariance and chamber/diffusion measurements conducted at
the same time. These observations might not have identical site coordinates but they are often
very close to each other (<500 m away from each other). Including multiple methods from the
same site provides an opportunity to compare estimates from different methods over a larger
number of sites.






## 4.2. Uncertainties in eddy covariance flux partitioning

Monthly Reco and GPP fluxes derived from eddy covariance were primarily estimated using flux partitioning based on night-time Reco based on the assumption that during night, NEE measured at low PAR is equivalent to Reco (Reichstein et al., 2005). Focusing on night-time partitioning ensured that data from older sites using this partitioning method could be included, and that most of the fluxes were standardized using one common partitioning method. However, particularly at sites at higher latitudes of the ABZ, low-light night-time conditions are restricted to rather short periods during summer, limiting the database for assessing Reco rates and therefore increasing uncertainties associated with the night-time partitioning (López-Blanco et al., 2020). Recent research suggests that other methods such as daytime partitioning (Lasslop et al. 2010), and even more recently artificial neural networks (ANN) (Tramontana et al., 2020), might be more accurate methods for flux partitioning by addressing the assumptions from night-time partitioning methods (Pastorello et al., 2020; Papale et al., 2006; Reichstein et al., 2005; Keenan et al., 2019). Specifically, the assumption of a constant diel temperature sensitivity during night- and daytime might introduce error in eddy covariance-based Reco estimates extrapolated from night-time measurements (Järveoja et al., 2020; Keenan et al., 2019). It should be noted that ABCflux database used night-time partitioning of fluxes extracted from repositories for consistency; however, fluxes contributed by some databases, PIs or extracted from papers may be based on other partitioning methods, as noted in the database. In a few cases, observations from the same site were based on different partitioning methods, which limits the usage of data at those sites for time-series exploration. These different gap-filling and partitioning approaches can impact the magnitude of monthly $CO_2$ budgets. For example, a study comparing four gap-filling methods in a boreal forest showed that the 14-year average annual NEE budget varied from 4 to 48 g C m$^{-2}$ year$^{-1}$ depending on the gap-filling approach (Soloway et al., 2017). However, a comparison of multiple gap-filling and partitioning methods across sites showed that variation in annual GPP and Reco between partitioning methods was small (Desai et al., 2008), which provides confidence in estimates from partitioned GPP and Reco components from the differing methods used in this database.







We suggest data users remain cautious when using ABCflux data to understand mechanistic
relationships between meteorological variables and fluxes, as the gap-filled and partitioned
monthly fluxes already include some information about, for example, air or soil temperatures and
light conditions. To completely avoid circularity in these exploratory analyses, we recommend
data users download the original and non-gap filled NEE records, or download fluxes partitioned
in a way that is consistent and biologically relevant for the particular research question from flux
repositories.

4.3. Representativeness and completeness of the data
The ABCflux database site distribution covers all vegetation types and countries within the ABZ.
However, there are regional and temporal biases in the database due to the differences in
accessibility for sampling certain regions (also documented in (Virkkala et al., 2019)). As a
result, the number of monthly observations does not always correlate with the size of the
country/region or vegetation type. For example, Russia and Canada cover in total ca. 80 % of the
ABZ but include only ca. 40 % of the monthly observations. While the distribution of these
measurements seems to be rather balanced between the Russian tundra and boreal biomes,
Canadian observations are primarily located in the boreal biome, largely due to the high amount
of measurements conducted as part of the Boreal Ecosystem-Atmosphere Study (Sellers et al.,
1997). Deciduous needleleaf (i.e., larch) forests, the primary vegetation type in central and
eastern Siberia, has the smallest amount of data compared to its area (<5 % of monthly
observations vs. >20 % coverage of the ABZ). Additional data gaps are located in barren and
prostrate-shrub tundra and sparse boreal vegetation. Eddy covariance towers in mountainous
regions are also rare as eddy covariance towers are most often set up over homogeneous and flat
terrains to avoid advection (Baldocchi, 2003; Etzold et al., 2010). Alaska and Finland cover <10
% of the ABZ but include >40 % of the monthly observations. Sites with NEE observations have
the largest geographical coverage, with less availability for partitioned GPP and Reco fluxes.
Therefore, regional summaries of Reco and GPP do not sum up to NEE. Moreover, although the
oldest records in ABCflux originate from 1989, observations from the 1990s are primarily
located in a few boreal or Alaskan tundra sites. The measurement records from tundra sites are



shorter than boreal sites over the full time span of the database, and it is therefore more uncertain
to investigate long-term temporal changes in tundra fluxes. Finally, the lowest amount of flux
data in ABCflux is from winter, which is the most challenging period for data collection in high
latitudes (Kittler et al., 2017b; Jentzsch et al., 2021). Fluxes are generally small during this
period (Natali et al., 2019), leading to higher relative uncertainties in flux estimation compared
to other seasons. These regional and temporal biases need to be considered in future analyses to
assure the robustness of our understanding of C fluxes across the ABZ.

Although ABCflux includes a comprehensive compilation of flux and supporting environmental
and methodological information, the information is not exhaustive. We acknowledge that this
database is missing some eddy covariance sites that were recently summarized in a tower survey
(see preliminary results in https://cosima.nceas.ucsb.edu/carbon-flux-sites/), because these data
were unavailable at the time of database compilation. Moreover, the overall quality or the gap-
filled percentage of the eddy covariance observations is not reported for each eddy covariance
site, limiting the potential to explore the effects of data quality on fluxes across all the eddy
covariance sites. Comparing soil temperature or moisture across sites has uncertainties due to
differences in sensor depths, which are not always reported in the database. We hope to improve
and increase the flux and supporting data in the future as new data are being collected, for
example, by leveraging the ONEflux pipeline and its different outputs (Pastorello et al., 2020), as
well as aggregating new measurements that are not part of any networks.
5. Data use guidelines
Data are publicly available using a Creative Commons Attribution 4.0 International copyright
(CC BY 4.0). Data are fully public, but should be appropriately referenced by citing this paper
and the database (see Section 6). We suggest that researchers planning to use this database as a
core dataset for their analysis contact and collaborate with the database developers and relevant
individual site contributors.
6. Data availability and access
The database associated with this publication can be found at Virkkala et al. 2021a
(https://doi.org/10.3334/ORNLDAAC/1934).



## 7. Conclusions

ABCflux provides the most comprehensive database of ABZ terrestrial ecosystem $CO_2$ fluxes to date. It is particularly useful for future modeling, remote sensing, and empirical studies aiming to understand $CO_2$ budgets and regional variability in flux magnitudes, as well as changes in fluxes through time. It can also be used to understand how different environmental conditions influence fluxes, and to better understand the current extent of the flux measurement network and its representativeness across the Arctic-Boreal region.

## 8. Author contributions

The ABCflux database was conceptualized and developed by a team led by SMN, BMR, JDW, MM, AMV, and EAGS, with additional comments from OS. KS and SJC compiled the data, with contributions from AMV, MM, DP, CM, and JN, and data screening by AMV and SMN. AMV drafted and coordinated the manuscript in close collaboration with SMN, BMR, JDW, KS, and MM. All authors contributed to the realization of the ABCflux database and participated in the editing of the manuscript. PIs whose data were extracted from publications are not coauthors in this paper, unless new data were provided, but their contact details can be found in the database.

## 9. Competing interests

The authors declare that they have no conflict of interest.

## 10. Acknowledgements

AMV, BMR, SMN, and JDW were funded by the Gordon and Betty Moore Foundation (grant #8414). BMR, KS, SJC, CM, and JN were also funded by the NASA Carbon Cycle Science and Arctic-Boreal Vulnerability Experiment programs (ABoVE grant NNX17AE13G), SMN by NASA ABoVE (grant NNX15AT81A) and JDW by NNX15AT81A and NASA NIP grant NNH17ZDA001N. EAGS acknowledges NSF Research, Synthesis, and Knowledge Transfer in a Changing Arctic: Science Support for the Study of Environmental Arctic Change (grant #1331083) and NSF PLR Arctic System Science Research Networking Activities (Permafrost Carbon Network: Synthesizing Flux Observations for Benchmarking Model Projections of Permafrost Carbon Exchange; grant #1931333. EAGS further acknowledges US Department of



Energy and Denali National Park. MBN and MP acknowledge Swedish ICOS (Integrated Carbon
Observatory System) funded by VR and contributing institutions; SITES (Swedish Infrastructure
for Ecosystem Science) funded by VR and contributing institutions; VR (grant # 2018-03966 and
# 2019-04676), FORMAS (grant # 2016-01289), and Kempe Foundations (SMK-1211). EE, CE,
and MSB-H was funded by NSF Arctic Observatory Network and CAG, VSL, EH by Natural
Sciences and Engineering Research Council. IM, PK, EST, AL acknowledges ICOS-Finland and
AV Russian Science Foundation, project 21-14-00209. AL, MA, TL, J-PT, and JH further
acknowledge Ministry of transport and communication. WQ, EE, VSL were funded by
ArcticNet. HK acknowledges The Arctic Challenge for Sustainability and The Arctic Challenge
for Sustainability II (JPMXD1420318865), MEM the Academy of Finland project PANDA
(decision no. 317054) and CV the Academy of Finland project MUFFIN (decision no. 332196).
NN acknowledges Arctic Data Center, National Science Foundation, US Department of Energy,
Denali National Park.YM was funded by Ministry of Environment, Japan and MU by the Arctic
Challenge for Sustainability II (ArCS II; JPMXD1420318865) and KAKENHI (19H05668).
SFO acknowledges US National Science Foundation, and MiM, BE, TRC Greenland Ecosystem
Monitoring program. BE further acknowledge Arctic Station, University of Copenhagen and the
Danish National Research Foundation (CENPERM DNRF100). ELB was funded by "Greenland
Research Council, grant number 80.35, financed by the "Danish Program for Arctic Research",
and LM by TCOS Siberia. DH and LK were funded by Deutsche Forschungsgemeinschaft under
Germany's Excellence Strategy – EXC 177 'CliSAP - Integrated Climate System Analysis and
Prediction'. JJ acknowledges Swedish Forest Society Foundation (2018-485-Steg 2 2017) and
FORMAS (2018-00792). DZ was funded by National Science Foundation (NSF) (award number
1204263, and 1702797) NASA ABoVE (NNX15AT74A; NNX16AF94A) Program, Natural
Environment Research Council (NERC) UAMS Grant (NE/P002552/1), NOAA Cooperative
Science Center for Earth System Sciences and Remote Sensing Technologies (NOAA-
CESSRST) under the Cooperative Agreement Grant # NA16SEC4810008, European Union's
Horizon 2020 research and innovation program under grant agreement No. 72789. S-JP was
funded by National Research Foundation of Korea Grant from the Korean Government (NRF-
2021M1A5A1065425, KOPRI-PN21011). NC acknowledges "National Research Foundation of
Korea Grant from the Korean Government (MSIT; the Ministry of Science and ICT) (NRF-
2021M1A5A1065679 and NRF-2021R1I1A1A01053870)". SD was funded by Department of





Energy and NGEE- Arctic. FJWP is funded by the Swedish Research Council (registration nr.
2017-05268) and the Research Council of Norway (grant no. 274711). The authors would like to
acknowledge Tiffany Windholz for her work on standardizing and cleaning up the database.

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
