# Peer review of "The ABCflux database: Arctic-Boreal CO2 flux observations and ancillary information aggregated to"

_Earth System Science Data, 2021_

## Author Comment (AC1)

**RC1: 'Comment on essd-2021-233', Anonymous Referee #1, 17 Sep 2021**

**Summary**

**The ABCflux dataset and companion manuscript provides, to my knowledge, the largest compilation of Arctic and boreal region carbon dioxide flux data, including net ecosystem exchange, and component ecosystem fluxes. This compilation therefore represents an unprecedented resource for synthesis studies aiming to understand high latitude carbon cycling and it's vulnerability to rapid high latitude global changes. The manuscript summarises the data acquisition and acquisition process undertaken and provides useful visualizations of the dataset that inform the reader of the main characteristics of the carbon exchanges, broken down by measurement approach, as well as dataset spatial and temporal coverage and representativeness. Overall the manuscript is well written and the dataset is comprehensive, logically structured, and carefully compiled, without any obvious errors. However, I have several minor-moderate remarks below about the manuscript and the dataset that should be addressed.**

**The main comment I have for this dataset and manuscript relates to pre-processing decisions, specifically the gap-filling methods for the eddy covariance data and the aggregation of the data to monthly fluxes, and the potential effect of these two decisions on uncertainties. Text describing these decisions and their effects on uncertainties is treated summarily in the manuscript text, or not at all, and therefore text should be strengthened/added as required. If a clear justification of these decisions cannot be provided, I think the dataset could be revised to include both a raw data file and a monthly-aggregated data file (the current version) to allow for more customization/fidelity for data users.**

Thank you for your positive feedback and for your important points concerning the gap-filling approaches and data aggregation decisions.

1) The gap-filling methods for the eddy covariance data

We added a new paragraph (2.2. Partitioning approaches at eddy covariance flux sites) describing our decision for the selected partitioning approach(es) and uncertainties related to it. We acknowledge that any one choice in partitioning introduces uncertainties. Future versions of ABCflux might strive to include fluxes calculated using multiple partitioning techniques, but by the nature of how we compiled the first version of this database it was beyond our scope.

Lines 378-402: "2.2. Partitioning approaches at eddy covariance flux sites
ABCflux compiles eddy covariance observations that were primarily partitioned using night-time Reco, which is based on the assumption that during night, NEE measured at low light levels is equivalent to Reco (Reichstein et al., 2005). This night-time partitioning approach has been the most frequently used approach to fill gaps in flux time series (Wutzler et al., 2018) due to its simplicity, strong evidence of temperature sensitivity of respiration, and direct use of Reco (i.e. night-time NEE) flux data to estimate temperature response curves (Reichstein et al., 2005). As the night-time approach was one of the first widely used partitioning approaches, fluxes partitioned with the approach were the only ones available in the flux repositories at some of the older sites. Daytime partitioning and other approaches started to develop more rapidly in the 2010s (Lasslop et al., 2010; Tramontana et al., 2020). Each of the partitioning approaches have uncertainties related to the ecological assumptions, input data, model parameters, and statistical approaches used to fill the gaps.

PIs that submitted data to us directly gap-filled and partitioned fluxes using the approach that they determined works best at their site. Based on similar logic, fluxes extracted from papers were not always partitioned using the night-time approach. In these cases, we trusted the expertise of PIs and authors, and included fluxes partitioned using other methods. Although this created some heterogeneity in the flux processing algorithms in the database, this approach was chosen so that we could be more inclusive with the represented sites.

Thus, in summary, our goal was to compile fluxes that 1) can be easily compared with each other (i.e., have been gap-filled and partitioned in a systematic way), 2) are as accurate as possible given the site conditions and measurement setup (i.e., other approaches were accepted if this was suggested by the PI), and 3) summarize information about the processing algorithms used."

Lines 780-791: "Any one choice in gap-filling and partitioning introduces uncertainties, and to understand and minimize those uncertainties remains an important research priority. However, since this database was not designed for detailed explorations of how the different gap-filling and partitioning approaches influence fluxes, we recommend users interested in those to access these data in flux repositories or contact site PIs. Fluxes calculated using multiple gap-filling techniques may be considered in the next versions of ABCflux. We further suggest data users remain cautious when using ABCflux data to understand mechanistic relationships between meteorological variables and fluxes, as the gap-filled and partitioned monthly fluxes already include some information about, for example, air or soil temperatures and light conditions. To completely avoid circularity in these exploratory analyses, we recommend data users download the original and non-gap filled NEE records, or download fluxes partitioned in a way that is consistent and biologically relevant for the particular research question from flux repositories."

2) The aggregation of the data to monthly fluxes

We appreciate the reviewers' comment and realize the limitations of any choice of standardized aggregation period. Ultimately, we derived monthly fluxes because our aim was to ensure our database could be easily used by as many potential users, and for as many important applications, as possible.  A majority of ecosystem process models, including those used in the Coupled Model Intercomparison Projects, as well as several remote sensing products (e.g., MODIS vegetation index product MOD13A3; Didan 2015) provide output at monthly intervals. Our database therefore allows for direct integration of these products (when, for example, assessing environmental drivers or spatially upscaling) as well as comparisons to these products (for example comparing fluxes to CMIP6 models). Furthermore, raw data files are rarely published together with scientific papers, which are one of our primary data sources. We were therefore restricted to an aggregated period when extracting these data. We learned through experience that extracting data at temporal periods finer than monthly was more challenging and uncertain. Additionally, we note that our monthly data product is provided at a finer temporal resolution than past synthesis efforts (McGuire et al., 2012; Belshe et al., 2013; Virkkala et al., 2021), which aggregated Arctic-boreal $CO_2$ fluxes to seasonal or annual time frames. Finally, we note that flux data derived from repositories already contains publicly-available raw data files, which users can access.

We added clarifications about these in the following line:

Lines 279-294: "ABCflux focuses on the area covered by the northern tundra and boreal biomes (>45 °N), as characterized in (Dinerstein et al., 2017), Fig. 2)), and compiles in-situ measured terrestrial ecosystem-level $CO_2$ fluxes aggregated to monthly time periods (unit: g C m-2 month-1). We chose this aggregation interval as monthly temporal frequency is a common, straightforward, and standard interval used in many synthesis, modeling studies, remote sensing products, and process model output (Didan, 2015; Natali et al., 2019a; Hayes et al., 2014). Furthermore, scientific papers often report monthly fluxes, facilitating accurate extraction to ABCflux. We compiled only aggregated fluxes to allow easy usage of the database, and to keep the database concise and cohesive. We designed this database so that these monthly fluxes, compiled from scientific papers or data repositories or contributed by site principal investigators (PIs), can be explored from as many sites as possible and across different months, regions and ecosystems. The database is not designed for studies exploring flux variability within a month, or how different methodological decisions (e.g., flux filtering or partitioning approaches) influence the estimated fluxes. If a potential data user requires fluxes at higher temporal frequency or is interested to study the uncertainties related to flux processing, we suggest they utilize data from other flux repositories (see Section 2.1.2.) or contact PIs."

**Comments on the Manuscript**

**249 - Could you justify more why the decision was made to aggregate to monthly timesteps as opposed to providing raw data and index columns that would allow for users to do their own aggregations.**

See our response above.

**Table 1 - I am not sure what is meant by "Soil respiration (or NEE)..." in the Natali et al. (2019) entry. Was only one of Rsoil or NEE ever used? Or is it more accurate use "and" as in the entries for the other studies?**

Thank you for this question, we changed it to "and".

**2.1.2 Flux repositories - I am confused about the processing pipeline for these tower data. This paragraph should be restructured to follow the steps in a linear sequence as much as possible. gap-filling is normally performed after USTAR filtering, but it comes first here. What is meant by "When only daily gap-filled data were supplied"? Aren't the data half-hourly? I also do not understand what the second-to-last sentence (line 349-350) means.**

We clarified this section and added details about the Natali et al. 2019 entries to the previous section so that the second-to-last sentence can more easily be understood. The paragraph now describes the processing steps the following way in lines 350-369:

"We downloaded eddy covariance and supporting environmental data products from AmeriFlux (Novick et al., 2018), Fluxnet2015 (Pastorello et al., 2020), EuroFlux database cluster (ICOS, Carbon Extreme, Carbo Africa, GHG Europe, Carbo Italy, INGOS) (Paris et al., 2012; Valentini, 2003), and Station for Measuring Ecosystem-Atmosphere Relations (Hari et al., 2013). Data that were filtered for USTAR (i.e., low friction velocity conditions) and gap-filled were downloaded from repositories in 2018-2020. USTAR varied among sites due to differing site-level assumptions. We downloaded only gap-filled data that met the USTAR criteria for either the tower PI or given through the database processing pipeline. However, Fluxnet2015 provides several different methods for determining data quality based on different USTAR criteria. In this case, we used the Fluxnet2015 common USTAR threshold (CUT, i.e. all years at the site filtered with the same USTAR threshold (Pastorello et al., 2020)). For observations extracted from EuroFlux, USTAR thresholds for each site were derived as described in (Papale et al., 2006; Reichstein et al., 2005) using night-time data. We extracted fluxes readily aggregated to monthly intervals by the data processing pipeline from Fluxnet2015 and EuroFlux. These aggregations were not given in AmeriFlux and SMEAR. We downloaded daily gap-filled data from these repositories and summed the data to monthly time steps. We did not aggregate any repository GPP, Reco, or NEE datasets that were not gap-filled. If fluxes were available for the same site and period both in Natali et al., (2019b) and flux repository extractions, the flux repository observations were kept in the database. Some repositories supplied eddy covariance data version numbers, which were added to the flux database."

**2.1.3 solicitation - what type of data were these?**

These were both eddy covariance and chamber data. We clarified this in the text.

**Line 373 - "needed to be filled"?**

We corrected this.

**2.2 - There is no information on quality screening for chamber measurements. Can it be assumed that the published values are reliable?**

We clarified this in lines 425-442:

"At chamber and diffusion sites, we disregarded observations including a low number of temporal replicates within a month (<3 individual measurements in summer months) and only one measurement month to ensure the temporal representativeness of the measurements. For the spring (March-May), autumn (September-November), and winter (December-February) months, one temporal replicate was accepted due to scarcity of measurements outside the summer season (June-August); measurement frequency is included in the database. We excluded monthly summertime measurements with <3 temporal replicates because within summer months, meteorological conditions and the phenological status of the ecosystem can vary significantly (Lafleur et al., 2012; Euskirchen et al., 2012; Schneider et al., 2012; Heiskanen et al., 2021), and a single measurement is unlikely to capture this variability. Our decision to exclude measurements that have only one measurement month was based on our goal to assess the temporal variability of fluxes. We justified the acceptance of a lower number of temporal replicates for the other seasons based on the assumption that flux variability is lower during the winter months, and at least during most of the spring and autumn months, due to the insulating effects of snow (Aurela et al., 2002; Bäckstrand et al., 2010). We estimate that excluding measurements with <3 temporal replicates during the summer months resulted in a 10 % loss of data. In total, 98 % of the chamber observations were from published studies; we assume that the peer review process assessed the quality of published data."

**Fig 5. Caption is not accurate. Should be a letter for each panel**

We corrected this.

**Table 4 - This table is bit lacking. Separating the flux from uncertaintines makes it hard to read, and the component fluxes can be computed from each other, and thus not terribly informative.**

Thank you for pointing this out. We added the standard deviation behind the mean flux values. Some sites only measure NEE and consequently, NEE summaries might not entirely match with GPP and Reco statistics, which is why we decided to keep all the three fluxes in the table. We added a clarification about this to the table caption. If the reviewer feels important information is still missing from this table, we would appreciate further clarification.

**4.1 639-666: I think something that has not been addressed completely is the fact that to compute a daily or monthly aggregated flux from a few chamber measurements one has to not only aggregate, but also upscale significantly more in the temporal domain than compared to EC, which likely has more temporal coverage. A chamber measurement for one half hour may agree closely with an EC measurement of for the same half hour, or perhaps for some period of that day, or perhaps even for that month. However, surely the uncertainty around the upscaled chamber flux must be much larger than the EC aggregation which may have a large number of temporal replicates and is an integration of a larger area? I would like to see this issue expanded upon in this section.**

This is a great comment. We fully agree that this point needs to be addressed. We therefore added this discussion and combined it with existing chamber uncertainty discussion in lines 716-723:
"Furthermore, uncertainty around gap-filled monthly chamber fluxes is presumably larger than that of the eddy covariance because of the low temporal replication of chamber measurements. Manual chamber measurements might, for example, be conducted during a limited period which does not cover the range of meteorological and phenological conditions within a month. Additional uncertainties in chamber measurements include, for example, accurate determination of chamber volume, pressure perturbations, temperature increase during the measurement, and collars disturbing the ground and causing plant root excision."

**4.2 - 698-730: Building on my earlier comment about the details of the post-processing of EC data (in particular gap-filling choices). I wonder here why the filtered (but not**

**gap-filled) flux columns were not provided alongside the gap-filled and partitioned columns? It seems this would enable quick comparisons for various topics of concern. For instance, how does gap-filling affect monthly aggregations? How much does gap-filling affect mechanistic conclusions of modeling exercises?**

Please see our response to the major comments.

**4.3 Representativeness - the discussion of geographic bias is useful, as is the comment about biome coverage. However, there is not much detailed consideration of the coverage with respect to the environmental covariates measured. A representativeness analysis like that of the following could be beneficial:**

**Hoffman, F. M., Kumar, J., Mills, R. T., & Hargrove, W. W. (2013). Representativeness-based sampling network design for the State of Alaska. Landscape Ecology, 28(8), 1567–1586. https://doi.org/10.1007/s10980-013-9902-0**

**Delwiche, K. B., Knox, S. H., Malhotra, A., Fluet-Chouinard, E., McNicol, G., Feron, S., et al. (2021). FLUXNET-CH4: a global, multi-ecosystem dataset and analysis of methane seasonality from freshwater wetlands. Earth System Science Data, 13(7), 3607–3689. https://doi.org/10.5194/essd-13-3607-2021**

This is an interesting suggestion. We agree that a more detailed representativeness assessment might be useful for the community, but developing such an analysis would include several methodological decisions on which of the available approaches and gridded datasets to use that might not necessarily be in the scope of a data description paper. Those approaches are, for example, analyses based on Euclidean distances (Hoffman et al. 2013; Pallandt et al. 2021), machine learning model classification (Virkkala et al. 2019; Villarreal and Vargas 2021), or exploring the environmental space with simpler scatter plots (Metcalfe et al. 2018; Delwiche et al. 2021). Moreover, such a paper for the Arctic-Boreal eddy covariance flux network is currently in review in Biogeosciences by some of the authors of this paper (Pallandt et al. 2021), which we now cited in this paper too. However, to provide some more context about the environmental space covered by the sites, we added a plot showing the distribution of sites along mean annual temperature and precipitation gradients (i.e. the Whittaker plot), which is

often used in other synthesis database publications (e.g., Bond-Lamberty et al. 2020; Lembrechts et al. 2020). The following additions were made:

Lines 508-510: "We further used TerraClimate annual and seasonal air temperature and precipitation layers averaged over 1989-2020 to visualize the distribution of monthly observations across the Arctic-Boreal climate space (Abatzoglou et al., 2018)."

Lines 563-565: "The sites in ABCflux cover the most frequent climatic conditions across the Arctic-Boreal zone relatively well; however, conditions with high precipitation are lacking sites (Fig. 5)."

Lines 821-823: "Autumn and winter data included in ABCflux further covers a smaller Arctic-Boreal climate space, with no data coming from extremely cold or wet conditions (Fig. 5)."

[Figure]

Fig 5. Mean annual air temperature and precipitation conditions across the Arctic-Boreal zone (a), the entire ABCflux (b), and the air temperature and precipitation conditions across the different climatological seasons included in ABCflux (c-f). Arctic-Boreal climate space was defined based on a random sample of 20000 pixels across the domain.

**Comments on the Dataset:**

**The number of observations does not match data description. I assume this is because the observation 'unit' referred to in the text is not a month/site combination, but rather the flux/month/site combination? I think it would be better to report the month/site combination and describe how much of those month/sites have each component flux of interest. Especially since the unique ID (first column) refers to a month/site combo).**

Thank you for pointing this out. If you are referring to the number of observations in the database downloaded from ORNL DAAC, this was an issue from the ORNL DAAC site. For some reason, the repository only exported a database with the first ~1000 rows of the database. This should now have been fixed.

**Can you explain more the lack of data citations for 30% of the observations?**

Observations that remain unpublished lack data citation (these are often completely new sites). Moreover, some sites did not include any data citations at flux repositories.

**I noticed none of the `data_maturity` is "preliminary" or "reprocessed" why are these provided? For future database expansion?**

Yes. We clarified this in the table.

**Why is measurement month named `Interval`? That is not intuitive.**

This is a good point. We changed it to "interval_month".

**Could `Measurement_frequency` be changed to provide the exact number of observations aggregated for the month? Instead, it could be named `Measurement_count`. Additionally a column for `gap-_count` could be provided and grouped with `gap-_perc` column. I think this would provide more useful information and data for dataset manipulation.**

We thank the referee for this comment. Unfortunately quantifying this information is not feasible for us at the moment, given the large amount of time that would be required. We built the first version of this database largely based on the Natali et al. 2019 non-growing season database, where high-frequency data were summarized as >100. Furthermore, the eddy covariance flux processing pipelines that aggregate fluxes to monthly intervals, which were used as a main data source for flux repository data, do not directly provide this information. Therefore, we would need to go back to the half-hourly data to estimate this. Moreover, this information is rarely available in publications. However, we agree that creating those three columns would be highly

useful for the data user, and hope to include those to our next version of the database. For more details, see our next response.

**`gap-_perc`: why is there only 17% coverage for this variable? Shouldn't it at least be the same as the next variable (Tower_QA_QC.NEE.flag)?  This also raises the question of how to interpret aggregations from sparse chamber measurements (i.e. are you effectively gap-filling?). I assume the real gap-filling is only done for EC, therefore, somewhere you should make a clearer distinction between your methods and assumptions between EC and chamber aggregations.**

Thank you for this good question. These columns indeed are related to the gaps of the eddy covariance data alone. Measurement_frequency is the column that should be used to estimate the extent of gaps in chamber and diffusion sites. This information is unfortunately rarely included in papers or was only occasionally contributed by PIs. We clarified these descriptions in Table 2 and added more details related to the data sources for the QAQC flag and gap_perc in lines 405-423:

"Repository eddy covariance data were processed and quality checked using quality flags associated with monthly data supplied by the repository processing pipeline. Fluxnet2015 and EuroFlux database include a data quality flag for the monthly aggregated data indicating percentage of measured (quality flag QC = 0 in FLUXNET2015) and good-quality gap-filled data (quality flag QC = 1 in FLUXNET2015; average from monthly data; 0=extensive gap-filling, 1=low gap-filling); for more details see Fluxnet2015 web page (https://fluxnet.org/data/fluxnet2015-dataset/variables-quick-start-guide/) and (Pastorello et al., 2020)). Note that this quality flag field for the aggregated data differs from the ones calculated for half-hourly data derived directly from eddy covariance tower processing programs (such as Eddypro). We removed monthly data with a quality flag of 0. Data with quality flags >0 were left within the database for the user to decide on additional screening criteria. Note that the monthly data produced by the repository processing pipeline do not include separate gap-filled percentages or errors of model fit for NEE similar to those associated with the half-hourly data. However, we included these fields to the database as PIs contributing data or scientific papers sometimes had this information; however these fields were not used in data quality screening. Both the monthly quality flag and gap-filled percentage fields describe the amount and quality of

the gap-filled data that needed to be filled due to, for example, instrument malfunction, power shortage, extreme weather events, and periods with insufficient turbulence conditions."

**`Tower_QA_QC.NEE.flag` variable is confusing. It seems to involve both the amount of gap-filled data and the quality of the gap-filing. Please provide a clearer explanation how to interpret the value between 0 and 1.**

We clarified this following the suggestion of the other referee. See our response above for the clarification.

**`Method_error_NEE_gC_m2` why does this variable only have 23% coverage, when the NEE aggregations are for 91% of the dataset? More information needs to be provided about under what circumstances it was deemed possible to compute an error and why. It seems to me that it may be possible to estimate an error or an uncertainty for any aggregation (chamber or EC tower).**

This information is unfortunately rarely included in papers or was almost never contributed by the PIs. Further, monthly flux aggregations produced by the repository processing pipelines, which were our main data source, did not calculate such aggregated method error estimates. We compiled this information whenever possible but acknowledge the fact that we are missing this information for many rows. Therefore, the main purpose of this column is to give the data user a rough idea of the errors associated with the models used to gap-fill NEE.

**Finally, can the authors please justify why they did not include any standardized variables extracted from geospatial products to make the dataset more ready for use? Things like MAT, MAP, and elevation could easily be filled using the WorldClim and GeoMorpo products respectively.**

There is a large amount of different climate (e.g., TerraClimate, WorldClim) or digital elevation model datasets (e.g., SRTM-DEM, MERIT DEM) that could be used for this purpose. However, each of those have their own strengths and limitations (see e.g., (Cao et al. 2020)), and including data extracted from these layers would require detailed explanations of why one

product was chosen over another one. Therefore, we decided to focus on site-level data, and let users decide which ancillary data sources to integrate for their specific analyses.

---

## Author Comment (AC2)

**RC2: 'Comment on essd-2021-233', Anonymous Referee #2, 02 Oct 2021**

**General Comments:**

**This manuscript describes a new database (ABC Fluxes) of CO2 flux measurements in arctic and boreal ecosystems. Overall the manuscript is well written and clear. However, the process of downloading data needs to be clarified – as I explain below, the big green "Download Data" button on the ORNL DAAC website did not give me a complete data file. One apparently needs to scroll down to the bottom to request the entire dataset, but this is not at all apparent at first pass and could lead to users missing data.**

**I also question the decision to exclude studies with limited measurements during the summer (while including limited measurements in off-seasons due to data scarcity during those time periods). While I agree that data sets with limited repetition are more uncertain than data with many repetitions, ideally ABC Fluxes users could make their own judgements about whether to include this data in their work. I recognize that going back to include this data is likely unfeasible at this point, so please instead provide an idea of how many studies were excluded. Then, perhaps future versions of ABC Fluxes could include these additional studies.**

1) Data downloading

Thank you for this important feedback. We contacted ORNL DAAC in September and they have fixed the data downloading process now.

2) Excluding studies with limited measurements

We agree with the referee that our decision to exclude summertime monthly measurements with <3 temporal replicates might have led to a loss of data that could have been useful for some data users, and hope to add those to the next version of the database. However, we think that

there are some good justifications on why such measurements were excluded from the first version of the database which we aimed to clarify in lines 425-442:

"At chamber and diffusion sites, we disregarded observations including a low number of temporal replicates within a month (<3 individual measurements in summer months) and only one measurement month to ensure the temporal representativeness of the measurements. For the spring (March-May), autumn (September-November), and winter (December-February) months, one temporal replicate was accepted due to scarcity of measurements outside the summer season (June-August); measurement frequency is included in the database. We excluded monthly summertime measurements with <3 temporal replicates because within summer months, meteorological conditions and the phenological status of the ecosystem can vary significantly (Lafleur et al., 2012; Euskirchen et al., 2012; Schneider et al., 2012; Heiskanen et al., 2021), and a single measurement is unlikely to capture this variability. Our decision to exclude measurements that have only one measurement month was based on our goal to assess the temporal variability of fluxes. We justified the acceptance of a lower number of temporal replicates for the other seasons based on the assumption that flux variability is lower during the winter months, and at least during most of the spring and autumn months, due to the insulating effects of snow (Aurela et al., 2002; Bäckstrand et al., 2010). We estimate that excluding measurements with <3 temporal replicates during the summer months resulted in a 10 % loss of data. In total, 98 % of the chamber observations were from published studies; we assume that the peer review process assessed the quality of published data."

We added the following description of studies that were seasonal in lines 330-340:

"We did not include fluxes reported at longer timesteps (e.g., seasonal aggregations), which, based on our rough estimate, resulted in a 10-20 % loss of data from sites and periods that would have been new to ABCflux. These excluded data primarily included some older, non-active eddy covariance sites and seasonal chamber measurements (e.g., (Nobrega and Grogan, 2008; Heliasz et al., 2011; Fox et al., 2008)). However, many of these data were located in the vicinity of existing sites covered by ABCflux (e.g., Daring Lake, Abisko), thus excluding these measurements does not dramatically influence the geographical coverage of the sites."

We want to stress that studies that measure fluxes to estimate monthly budgets often measure them more than once during the summer months; thus our decision to exclude measurements

with <3 temporal replicates during the summer months did not lead to significant losses of data. Studies that have fewer replicates often standardize fluxes to a common light level and temperature to compare across sites (e.g., NEE at 600 PAR or Reco at 20 Celsius) instead of trying to provide monthly mean fluxes or extrapolate the measurements across the entire month (see e.g., Shaver et al. 2007, Sorensen et al. 2019, Cahoon et al. 2016).

**Specific Comments:**

**Line 208: "fluxes from plants and soils to the atmosphere."**

We corrected this.

**Line 216: Since you have included the measurement scale for eddy covariance and chamber measurements, please comment on the measurement scale for snow diffusion.**

We added this information.

**Line 250: At some point in paper, please quantify (at least approximately) how much data you lose by using monthly values and excluding papers that do not report data on a monthly basis.**

See our response above.

**Lines 304 – 309: the sub-plot labels b, c, d, and e are mixed up**

We corrected this.

**Line 313-314: Here you only cite 5 prior synthesis efforts, but your Table has 7 other efforts.  Is this an oversight, or is there a reason you did not look at two of the synthesis papers to identify potential data?**

We added Baldocchi et al. 2018 and Luyssaert et al. 2007 to the citation.

**Line 323-328: I am curious to hear more about why you decided to exclude summer measurements if there weren't many replicates.  While fewer replicates will make the uncertainty higher, eliminating these data sets entirely could throw away potentially valuable information that ABCfluxes users may want.  It seems to me that, in an ideal world, ALL data of sufficient quality (if not quantity) would be included, and then database users can decide whether or not to include these small studies in their results. I am not suggesting you re-do the database now to include all these disregarded summer points, I recognize that would likely be a huge time investment, but I am curious to know approximately how many datasets were discarded.**

See our response above.

**Line 364-368:  Please clarify here that in FLUXNET2015, "_QC = 0" means measured values and "_QC = 1 means good quality gap-filled values.  Otherwise, your phrase "0 = extensive gap-filling, 1=low gap-filling" could be interpreted to conflict with the FLUXNET2015 QC designations and may confuse people.  You could write something like "indicating percentage of 366 measured (quality flag QC = 0 in FLUXNET2015) and good-quality gap-filled data (quality flag QC = 1 in FLUXNET2015); average from daily data; 0=extensive gap-filling, 1=low gap-filling).**

We added this correction, together with a reference to the Fluxnet2015 web page where this is described.

**Line 397: Ah!  I now see that the dataset is supposed to have 6309 rows.  The first time I downloaded this dataset I went to https://doi.org/10.3334/ORNLDAAC/1934, logged in, and clicked on the big green "Download data" button near the top of the page.  However, the file I get from this only has 1408 rows.  I now see that if I scroll down the webpage I**

**can request a much large file.  Why does the "download data" button not provide the full dataset?  Can this be changed?  If not, you may want to warn the user about this in your text.**

Thank you for pointing this out - we had not noticed it. ORNL DAAC should now have fixed it.

**Table 3: please clarify that by "Number of observations" you mean # of months of data.**

We clarified this.

**Line 515: Please clarify that the likely reason you have less data from 2015-present is because of a reporting lag, not because eddy covariance towers are measuring less data now.**

Thank you for this important point - we added a sentence about this.